# Robust genome editing activity and the applications of enhanced miniature CRISPR-Cas12f1

Soo-Ji Park [1,2,4], Sungjin Ju [1,2,4], Won Jun Jung[1,2], Tae Yeong Jeong [1,2], Da Eun Yoon [1,2], Jang Hyeon Lee[1,2], Jiyun Yang [1,2], Hojin Lee[2], Jungmin Choi [2], Hyeon Soo Kim[2,3] & Kyoungmi Kim [1,2] ✉

With recent advancements in gene editing technology using the CRISPR/Cas system, there is a demand for more effective gene editors. A key factor facilitating efficient gene editing is effective CRISPR delivery into cells, which is known to be associated with the size of the CRISPR system. Accordingly, compact CRISPR-Cas systems derived from various strains are discovered, among which Un1Cas12f1 is 2.6 times smaller than SpCas9, providing advantages for gene therapy research. Despite extensive engineering efforts to improve Un1Cas12f1, the editing efficiency of Un1Cas12f1 is still shown to be low depending on the target site. To overcome this limitation, we develop enhanced Cas12f1 (eCas12f1), which exhibits gene editing activity similar to SpCas9 and AsCpf1, even in gene targets where previously improved Un1Cas12f1 variants showed low gene editing efficiency. Furthermore, we demonstrate that eCas12f1 efficiently induces apoptosis in cancer cells and is compatible with base editing and regulation of gene expression, verifying its high utility and applicability in gene therapy research.

Clustered regularly interspaced short palindromic repeats (CRISPR)-based gene editing is a technology that precisely manipulates the genes in an organism through DNA deletion, insertion, or base replacement. It is a promising technology not only for the treatment of genetic diseases but also for the treatment of cancer caused by mutations that activate oncogenes[1,2].

CRISPR/Cas9 is a third-generation genome editing technology derived from a bacterial defense system[3]. It consists of Cas9, which exhibits DNA cleavage activity, and a chimeric single guide RNA (sgRNA) that localizes the Cas9-sgRNA complex to a specific genomic site. The DNA cleavage ability of Cas9 can be used for loss of gene function, and catalytically inactive Cas9 can be used with a variety of effector molecules, such as fluorescent proteins, gene regulatory factors, and deaminases for cell imaging, regulation of gene expression, and base editing[3].

Recently, Cas9 orthologs derived from various strains have been discovered, and CRISPR systems of various sizes have been reported[4]. Effective delivery of the CRISPR system to target cells and tissues is critical for successful gene editing, and the size of the CRISPR system has been cited as a factor that affects effective CRISPR delivery[5,6]. In this context, small CRISPR systems have attracted attention because they are advantageous for efficient delivery and can be easily applied to various delivery vehicles[6–8].

Un1Cas12f1 (hereinafter referred to as Cas12f1) originates from uncultured archaea and is categorized as a class 2 type V-F CRISPR RNA-guided enzymes. Cas12f1 (529 aa) is approximately 2.6 times smaller than SpCas9 (1368 aa) and AsCpf1 (1307 aa)[9] and an asymmetric Cas12f1 dimer (Cas12f1.1 and Cas12f1.2) assembles with the sgRNA to recognize and cleave the double-stranded DNA[10,11]. SpCas9 and AsCpf1 are widely used because of their high gene editing

[1]Department of Physiology, Korea University College of Medicine, Seoul, Republic of Korea. [2]Department of Biomedical Sciences, Korea University College of Medicine, Seoul, Republic of Korea. [3]Department of Anatomy, Korea University College of Medicine, Seoul, Republic of Korea. [4]These authors contributed equally: Soo-Ji Park, Sungjin Ju. ✉e-mail: kim0912@korea.ac.kr

efficiency; however, because of their large molecular sizes the clinical applications of these CRISPR systems are often hindered by the challenge of delivering them into cells[5,6]. In contrast, Cas12f1 is smaller than SpCas9 and AsCpf1, even when various factors are fused to Cas12f1[12]. Although compact Cas12f1 provides important advantages for in vivo gene therapy studies, the indel efficiency of canonical Cas12f1 is marginal in mammalian cells[12–14].

To investigate the potential of Cas12f1 as an effective gene therapy tool, one study used engineered Cas12f1 and sgRNA (CasMINI) for efficient gene activation[12], and another study improved the gene editing activity of native Cas12f1 by modifying sgRNA (Cas12f1_ge4.1)[13]. Other studies have also verified the gene editing efficiency of the combination of Cas12f1 nuclease from CasMINI and sgRNA from Cas12f1_ge4.1[14–16]. However, low editing activity was still observed, depending on the target gene, and this issue has limited the use of Cas12f1. To address this problem, we applied a new strategy and reviewed previously reported studies to develop Cas12f1 with high editing efficiency, even for gene targets where low gene editing activity was observed.

As a result, we develop enhanced Cas12f1 (eCas12f1) by introducing five mutations (D143R, T147R, G181R, K330R, and S426G) to the Cas12f1 endonuclease and utilizing sgRNA_S1b with increased RNA structural stability. eCas12f1 shows an improved indel frequency of more than 30% for most gene targets used in this study. Additionally, eCas12f1 shows similar gene editing efficiency to SpCas9 and AsCpf1. We also evaluate its potential as a cancer therapy tool by applying eCas12f1 to disrupt *PLK1* gene to reduce cell survival and *BRAF* gene with V600E mutation for targeted therapy. Finally, we demonstrate that efficient base editing and regulation of gene expression were possible using catalytically inactivated Cas12f1. Overall, we show that eCas12f1 enables robust gene disruption, efficient base conversion, and regulation of gene expression, thereby verifying its potential as a highly useful tool in genetic engineering and gene therapy.

## Results

### Construction of initial plasmid for development of advanced Cas12f1 variant
In a recent study, to induce efficient activation of gene expression using Cas12f1 and transcriptional activators, dCasMINI-VPR was developed by mutating amino acids in the DNA-binding pocket of catalytically inactive Cas12f1 and engineering sgRNA. dCasMINI-VPR has been shown to activate gene expression with high efficiency in mammalian cells[12]. In another study, to improve the gene editing activity of natural Cas12f1, which showed low gene editing efficiency in mammalian cells, Cas12f1_ge4.1 was developed by optimizing the sgRNA of Cas12f1 and increased the average indel frequency by 867-fold[13]. In the two studies, sgRNAs were engineered for both Cas12f1 variants (sgRNA design #2[12] and sgRNA_ge4.1[13]), and the sgRNA of Cas12f1_ge4.1 was shorter than that of dCasMINI-VPR (120-nt and 183-nt, respectively). Thus, to maximize the advantages of compact Cas12f1, we began our study using the sgRNA of Cas12f1_ge4.1. Meanwhile, the original sequence of Cas12f1 was used in CasMINI and the codon-optimized sequence was used in Cas12f1_ge4.1. Therefore, we optimized the codons of native Cas12f1 and investigated the differences in gene editing efficiency between Cas12f1 with the codon optimized sequence and those with the original sequence across six endogenous human genes. This included four genes (*NLRC4*, *CLIC4*, *HBB*, *COL8A1*) used in the assessment of Cas12f1_ge4.1 and two genes (*RNF2*, *VEGFA*) that has not been previously evaluated. The transfection efficiency of plasmid using lipid delivery in the HEK293T cell line was 96% (GFP-positive cells), and the transfected cell pool was used to analyze editing efficiency (Fig. 1a and Supplementary Fig. 1a). We confirmed that codon optimization of the Cas12f1 sequence led to increased gene editing efficiency in four out of six targets compared to the original sequence (Supplementary Fig. 1b). Accordingly, we

constructed Cas12f1_v1, a single vector encoding both human-codon optimized Cas12f1 fused with a bipartite nuclear localization signal (bpNLS) at the N-terminus, and sgRNA_ge4.1 (Fig. 1b).

We then assessed the gene editing efficiency of Cas12f1_v1 in six target genes, finding that Cas12f1_v1 demonstrated a similar indel efficiency to Cas12f1_ge4.1 (Supplementary Fig. 1c). Therefore, we decided to use Cas12f1_v1 as the initial version for implementing our strategies to enhance the gene editing efficiency of Cas12f1.

### Enhancing gene editing efficiency by improving the stability of sgRNA secondary structure
A recent study showed that misfolding of guide RNA (gRNA) can occur depending on the spacer sequence, which affects the gene editing activity of SpCas9[17]. To prevent gRNA misfolding, researchers introduced the 5′-GGAC(UUCG)GUCC-3′ hairpin sequence, an exceptionally stable and very common RNA hairpin[18], into the first hairpin of SpCas9 gRNA, locking the gRNA folding structure[17]. We hypothesized that gRNA misfolding might be more prevalent in Cas12f1 sgRNA due to its greater length relative to SpCas9 gRNA. Therefore, we investigated whether applying a highly stable hairpin to the sgRNA of Cas12f1 could improve the stability of sgRNA structure, thereby increasing gene editing efficiency. We focused on Stem 1 and Stem 3, excluding Stem 2, which interacts with nucleotides -7 to -2 of sgRNA, and Stem 4, which is formed by the simple connection of tracrRNA and crRNA (Fig. 1c). Given that modifications to the 5′-GGAC(UUCG)GUCC-3′ sequence led to a slight reduction in the genome editing efficiency of SpCas9[17], we applied the stable hairpin sequence in its original form. First, we designed two stems for Stem 1. One approach was to retain the loop sequence without modification, as A$^{-86}$ in the loop interacts with K330 of Cas12f1.2[10]; therefore, only the stem sequence of the stable hairpin was introduced by replacing U$^{-94}$:A$^{-82}$ and C$^{-93}$:G$^{-83}$ with G:C base pairs (sgRNA_S1a). The other approach involved an additional modification to the loop sequence by substituting A$^{-88}$ with U to incorporate the UUCG loop sequence of the stable hairpin (sgRNA_S1b) (Fig. 1c, d). We investigated the effect of the sgRNA modifications on six endogenous genes in HEK293T cells, and the transfected cells were analyzed after 72 hr post-transfection. Compared with Cas12f1_v1, Cas12f1_S1a and Cas12f1_S1b improved gene editing efficiency by an average of 1.09-fold and 1.26-fold, respectively (Fig. 1f, g).

Next, we modified Stem 3, as the loop sequence of Stem 3 was identical to that of the stable hairpin; therefore, only the stem sequences of UUUC at positions -43 to -40 and GAAA at positions -35 to -32 were replaced with GGAC and GUCC, respectively (sgRNA_S3) (Fig. 1c, e). We evaluated Cas12f1_S3 on six endogenous genes and found an average indel efficiency of 1.09-fold relative to that of Cas12f1_v1 (Fig. 1h, i). We then applied sgRNA_S1b and sgRNA_S3 together, however it did not further increase gene editing efficiency (Supplementary Fig. 2a). Additionally, we investigated the gene editing efficiency of Cas12f1_S1b on five gene targets (Supplementary Fig. 2b) and found a significant increase in gene editing efficiency with an average of 1.63-fold in eight out of eleven gene targets (Fig. 1f and Supplementary Fig. 2b). We calculated the free energy of the sgRNA secondary structure using the RNAeval web server[19] and confirmed that the free energy of sgRNA_S1b was lower than that of sgRNA_ge4.1, suggesting that the modifications applied to sgRNA_S1b increased the stability of the sgRNA structure[18] (Supplementary Fig. 2c, d).

### Enhancing gene editing efficiency by increasing the interaction between Cas12f1 nuclease and sgRNA spacer
The DNA-binding pocket within the Cas nuclease interacts with DNA and sgRNA, contributing to the accessibility of the target DNA and DNA cleavage of the Cas nuclease[20]. Xu et al.[12] developed dCasMINI-V4 by replacing the four amino acids (D143, T147, K330, and E528) in the DNA-binding pocket of dead Cas12f1, containing D326A and D510A, with arginine to enhance its binding affinity with nucleic acids.

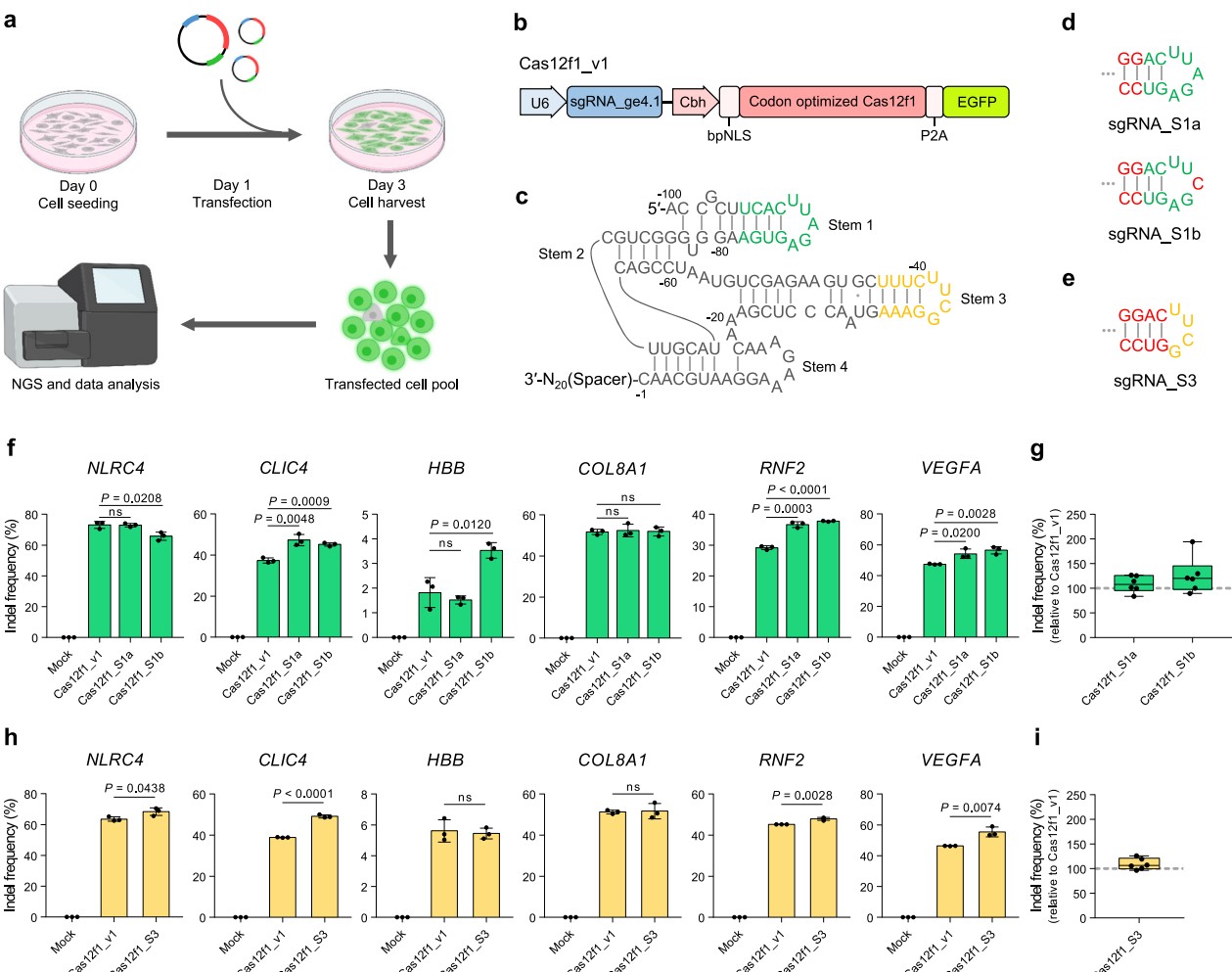

**Fig. 1 | Engineering Cas12f1 sgRNA to improve RNA structural stability.**
**a** Schematic illustration of the workflow to investigate the gene editing activity of Cas12f1. Created in BioRender. Kim, K. (2025) https://BioRender.com/s99u820 **b** Construction of Cas12f1_v1 comprising human-codon optimized Cas12f1 and sgRNA_ge4.1. **c** Structure of sgRNA_ge4.1. Stem 1 and Stem 3 are shown in green and yellow, respectively. **d** Modifications of Stem 1. Top: nucleotides of U$^{-94}$:A$^{-82}$ and C$^{-93}$:G$^{-83}$ pairs are replaced with G:C base pairs (sgRNA_S1a). Bottom: additional modification to sgRNA_S1a by replacing the loop sequence from UUAG to UUCG (sgRNA_S1b). Nucleotide changes are shown in red. **e** Modification of Stem 3. Nucleotides of UUUC at positions -43 to -40 and GAAA at positions -35 to -32 were replaced with GGAC and GUCC, respectively (sgRNA_S3). Nucleotide changes are

shown in red. **f** Comparison of indel frequencies of Cas12f1_v1, Cas12f1_S1a, and Cas12f1_S1b at six gene targets. **g** Relative indel frequencies of Cas12f1_S1a and Cas12f1_S1b at six gene targets from f ($n = 6$) with respect to the corresponding Cas12f1_v1, which is set to 100%. **h** Comparison of indel frequencies of Cas12f1_v1 and Cas12f1_S3 at six gene targets. **i** Relative indel frequencies of Cas12f1_S3 at six gene targets from h ($n = 6$) with respect to the corresponding Cas12f1_v1, which is set to 100%. Data in bar graphs represent the mean ± s.d. of three independent biological replicates. In the box-and-whisker plot, boxes represent the interquartile ranges with the median as a line, and whiskers extend from the minimum to the maximum values. *P*-values were obtained using the two-tailed Student's t-test. ns, not significant. Source data are provided as a Source Data file.

dCasMINI-V4, optimized with engineered sgRNA design #2, efficiently increased gene expression in mammalian cells. They also verified the gene cleavage activity of three CasMINI variants (CasMINI-V2 with D143R, T147R; CasMINI-V3.1 with D143R, T147R, E151A; and CasMINI-V4 with D143R, T147R, K330R, E528R) and the CasMINI-V3.1 showed the highest gene editing efficiency[12].

The editing efficiency of the combination of CasMINI nuclease and sgRNA_ge4.1 has been reported previously[14–16], but there has been no further optimization study for this combination. Additionally, due to differences in the scaffold structures between sgRNA design #2 and sgRNA_ge4.1, as well as spacer lengths (23 nt and 20 nt, respectively), we decided to re-examine the mutations introduced in CasMINI. We generated three Cas12f1_v1 variants (Cas12f1_mini2, Cas12f1_mini3.1, and Cas12f1_mini4) by introducing the corresponding mutations found in the CasMINI variants (Fig. 2a) and transfected them into HEK293T-cells. The analysis of gene editing efficiency in five gene targets, including a potential cancer therapy target (*PLK1*)[21], revealed that

Cas12f1_mini2 showed approximately 50% lower efficiency at two out of five target sites (*NLRC4* and *CLIC4*), while demonstrating an average of 2.88-fold increase in indel frequency at three out of five target sites (*HBB*, *PLK1*, and *VEGFA*) compared to Cas12f1_v1 (Fig. 2b). Cas12f1_mini3.1 exhibited an average indel efficiency of 2.14-fold across all five target sites compared to Cas12f1_v1. Cas12f1_mini4 showed similar gene editing efficiency to Cas12f1_v1 at two out of five target sites (*NLRC4* and *CLIC4*), while demonstrating an average of 3.28-fold increase in indel frequency at three out of five target sites (*HBB*, *PLK1*, and *VEGFA*) (Fig. 2b). Interestingly, Cas12f1_mini3.1, which was introduced with the same mutations as CasMINI_V3.1—the variant that showed the highest indel efficiency among the CasMINI variants—did not consistently exhibit higher indel efficiency compared to the other variants (Fig. 2b). This is likely due to CasMINI_V3.1 being optimized with sgRNA design #2 instead of sgRNA_ge4.1. To develop a Cas12f1 variant with optimal efficiency for sgRNA_ge4.1, we constructed Cas12f1_v2 by simply introducing all five mutations applied to the three

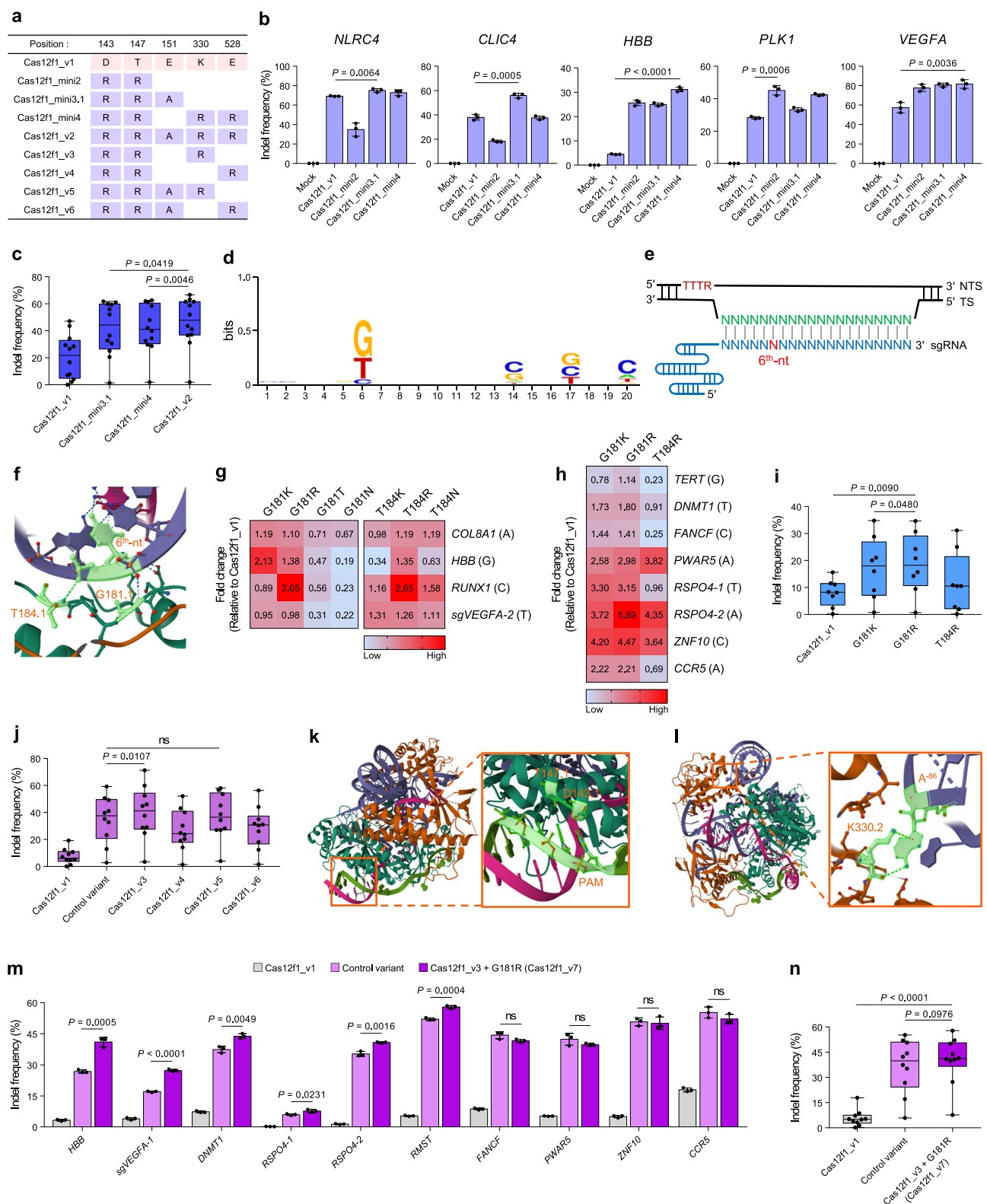

CasMINI variants (Fig. 2a) and compared its gene editing efficiency with Cas12f1_mini3.1 and Cas12f1_mini4 in twenty-eight target genes (Supplementary Table 1). The results showed that Cas12f1_v2 did not exhibit greater editing efficiency than Cas12f1_mini3.1 and Cas12f1_mini4 (Supplementary Fig. 3a), and only twelve out of twenty-eight targets showed higher indel frequency than the other variants (Fig. 2c). While exploring the common features of sgRNAs targeting the twelve genes, we found that the 6th nucleotide of the sgRNA spacer was predominantly guanine (G) or thymine (T) (Fig. 2d). Thus, we investigated whether there were amino acids that interacted with this 6th nucleotide. The previously characterized structure of Cas12f1 (PDB 7C7L)[10] revealed that G181 and T184 in Cas12f1.1 form hydrogen bonds with the backbone of the 6th nucleotide (Fig. 2e, f). We hypothesized that replacing these amino acids with positively charged or polar amino acids would enhance the interaction with the 6th nucleotide of the sgRNA spacer, thereby reducing nucleotide preference and

**Fig. 2 | Engineering Cas12f1 endonuclease to increase the interaction with target gene and sgRNA. a** Summary of Cas12f1 variants with different combinations of mutations in the DNA-binding pocket. **b** Comparison of indel efficiency among Cas12f1_v1, Cas12f1_mini2, Cas12f1_mini3.1, and Cas12f1_mini4 at five gene targets. **c** Indel frequencies of Cas12f1_v1, Cas12f1_mini3.1, Cas12f1_mini4, and Cas12f1_v2 at twelve targets. **d** Sequence logo of twelve target sites where Cas12f1_v2 showed higher indel frequency compared to Cas12f1_v1, Cas12f1_mini3.1, and Cas12f1_mini4. Logo was created by WebLogos[39]. **e** Position of the 6th nucleotide in the sgRNA spacer. TS, target strand; NTS, non-target strand; magenta letters, PAM sequence; light green letters, target sequence; light blue letters, sgRNA spacer sequence. **f** Location of G181 and T184 in Cas12f1.1 (PDB: 7C7L). G181.1, T184.1, and 6th nucleotide of the sgRNA spacer are highlighted in light green. **g** Heat map of relative indel frequencies of the seven mutated Cas12f1 variants at four gene targets. The seven variants contain different mutations in the G181 or T184 amino acid that interacts with the 6th nucleotide. The type of the 6th nucleotide is indicated in parentheses next to the gene name. **h** Additional experimental data for relative indel efficiency of G181K, G181R, and T184R variants at eight gene targets. The type of the 6th nucleotide is indicated in parentheses next to the gene name. **i** Comparison of indel frequencies of G181K, G181R, and T184R variants at eight gene targets from h (n = 8). **j** Comparison of indel frequencies of Cas12f1_v3 ~ 6 in ten gene targets where Cas12f1_v1 exhibited less than 20% editing efficiency. **k** Location of D143 and T147 in Cas12f1.1. D143, T147, and PAM are highlighted in light green. **l** Location of K330 in Cas12f1.2. K330 and A[-86] in Stem 1 of the sgRNA are highlighted in light green. **m, n** Comparison of indel efficiencies between the control variant, which exhibited higher editing efficiency for each target from Fig. 2j, and Cas12f1_v7 at ten gene targets. Data in the bar graphs represent the mean ± s.d. of three independent biological replicates. In the box-and-whisker plot, boxes represent the interquartile ranges with the median as a line, and whiskers extend from the minimum to the maximum values. *P*-values were obtained using the two-tailed Student's t-test. ns, not significant. Source data are provided as a Source Data file.

enhancing cleavage activity. To test this hypothesis, we created seven Cas12f1_v1 variants by substituting G181 or T184 with lysine, arginine, threonine, or asparagine. These variants were then evaluated for gene editing efficiency in four endogenous genes with different types of 6th nucleotides. The experimental results showed that the variants with G181K or G181R mutations exhibited an average increase in gene editing efficiency of 1.29-fold and 1.53-fold, respectively, compared to Cas12f1_v1 (Fig. 2g). In contrast, the G181T and G181N variants showed lower editing efficiency than Cas12f1_v1 across all targets. While T184K variant exhibited an average decrease in gene editing efficiency of 0.95-fold compared to Cas12f1_v1, T184N variant showed a slight increase of 1.13-fold, and T184R variant showed an average increase of 1.61-fold compared to Cas12f1_v1 (Fig. 2g). We further attempted to introduce mutations at both G181 and T184 simultaneously, but no noticeable improvement in gene editing efficiency was observed compared to the variants with a single mutation (Supplementary Fig. 3b). Next, we performed additional experiments on eight gene targets to select a variant with the greatest editing efficiency among the G181K, G181R and T184R variants. The results revealed that the G181K, G181R and T184R variants showed average increases in gene editing efficiency of 2.50-fold, 2.88-fold, and 1.86-fold, respectively (Fig. 2h). Consequently, the G181R variant exhibited significantly higher gene editing efficiency compared to the other variants (Fig. 2i).

We also introduced mutations in three amino acids interacting with the 15th nucleotide on the target strand, which is complementary to the 6th nucleotide of the spacer (Supplementary Fig. 3c, d). We constructed eleven Cas12f1 variants by substitution of the three amino acids with lysine, arginine, threonine, or asparagine. No variants showed a notable increase in editing activity compared to Cas12f1_v1 across all four gene targets, and the mutation at Y445 even abolished gene editing activity (Supplementary Fig. 3e).

**Improving gene editing efficiency via enhanced interaction between Cas12f1 nuclease and nucleic acids**
We further investigated mutations in the DNA-binding pocket because we speculated that there might be a more optimal combination of mutations for Cas12f1_v1. Thus, we constructed four additional Cas12f1 variants (Cas12f1_v3 ~ 6), each containing different combinations of mutations (Fig. 2a). The indel frequencies of the four Cas12f1 variants were assessed in ten targets (Source Data), where Cas12f1_v1 exhibited indel efficiencies of less than 20% among the twenty-eight targets used in Supplementary Fig. 3a. The experimental results showed that Cas12f1_v3, which includes D143R, T147R, and K330R mutations, exhibited greater indel efficiency compared to the control variants that showed highest indel efficiency at each target used in Supplementary Fig. 3a (Fig. 2j and Source Data). To investigate the effects of these mutations on DNA cleavage activity, we examined the structure of Cas12f1 (PDB 7C7L)[10]. Among the three mutations, D143R and T147R

in Cas12f1.1 were located near the protospacer adjacent motif (PAM) (Fig. 2k). We speculated that the substitution of negatively charged aspartic acid with positively charged arginine enhances electrostatic interactions between the Cas12f1 nuclease and DNA of PAM sequence[20]. Furthermore, we found that K330 in Cas12f1.2 interacts with A[-86] in Stem 1 of the sgRNA. We hypothesize that replacing lysine with arginine, which has a stronger positive charge, increases the electrostatic attraction between Cas12f1 nuclease and sgRNA, thereby enhancing the stability of the Cas12f1-sgRNA complex (Fig. 2l).

We combined the modifications applied to Cas12f1_S1b, G181R variant, and Cas12f1_v3. First, we introduced G181R mutation to Cas12f1_v3 to determine if the mutation had an additional impact on indel efficiency, and experiments were conducted on the ten gene targets used in Fig. 2j. As a control, we used Cas12f1 variants that exhibited the maximum efficiency for each gene target in Fig. 2j (control variant). In six out of ten targets, Cas12f1_v3 containing G181R mutation showed a significant additional increase in indel efficiency, with an average of 1.28-fold higher compared to the control variants (Fig. 2m). For the remaining four targets, no significant difference in indel efficiency was observed (Fig. 2m). Based on these results, introducing the G181R mutation to Cas12f1_v3 resulted in similar or increased gene editing efficiency relative to the control variant (Fig. 2m, n).

Next, we applied sgRNA_S1b to the Cas12f1_v3 + G181R variant. Evaluating it on the same ten gene targets, we observed a slight increase in editing efficiency with an average of 1.16-fold in only three out of ten targets and decreased indel efficiency in two targets relative to the Cas12f1_v3 + G181R variant (Supplementary Fig. 4a). The application of the highly stable hairpin in Stem 1 did not effectively increase the gene editing efficiency of the Cas12f1_v3 + G181R variant (Supplementary Fig. 4a, b). Therefore, we termed Cas12f1_v3 with G181R variant as Cas12f1_v7 (Fig. 2m, n).

**Development of enhanced Cas12f1 with robust gene editing activity**
Recently, two studies reported the engineered AsCas12f (422 aa), a family member of the type V-F Cas12f derived from *Acidibacillus sulfuroxidans*[16,22]. In one study, the amino acid sequences of AsCas12f and AsCas12f homologs were aligned to identify positions where AsCas12f contains neutral or negatively charged residues, while its homologs have positively charged residues. Selected amino acids in AsCas12f were then replaced with positively charged residues, and ultimately, five mutations (D196K, N199K, G276R, N328G, and D364R) were introduced, resulting in the development of an enhanced AsCas12f (enAsCas12f) with improved DNA targeting and cleavage activity[16]. In another study, deep mutational scanning (DMS) was performed to investigate the effects of all 20 amino acid substitutions in the entire AsCas12f sequence, which led to the development of two

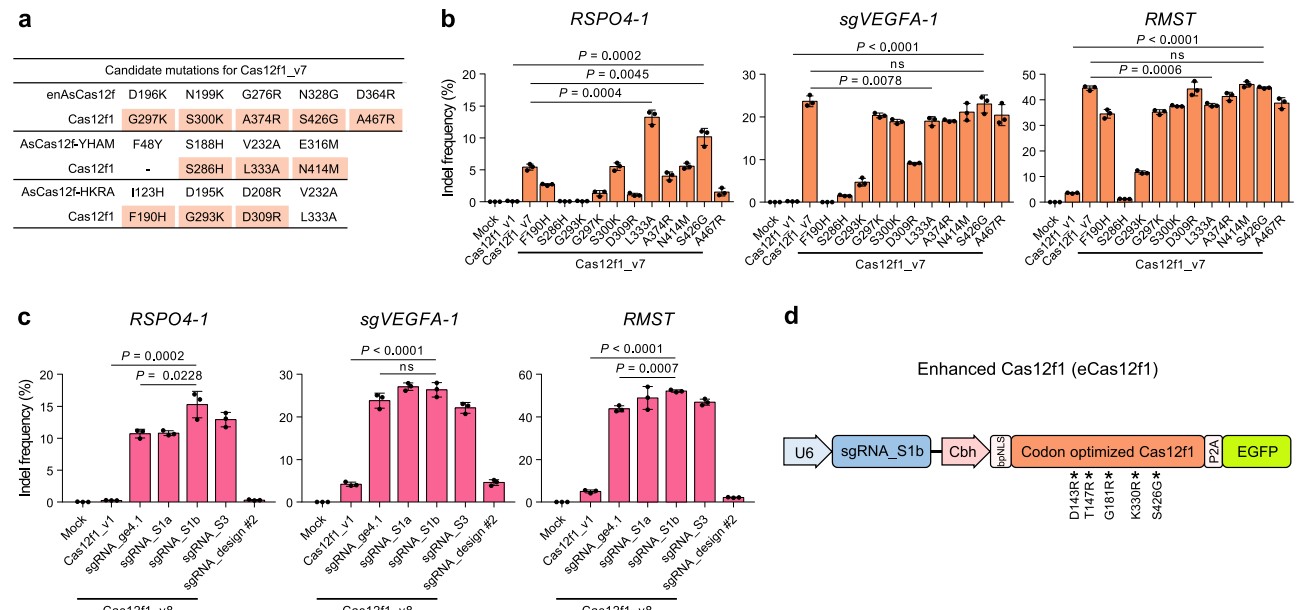

**Fig. 3 | Development of enhanced Cas12f1. a** Mutations introduced to enhance the gene editing efficiency of AsCas12f and the corresponding Cas12f1 amino acids from the sequence alignment. **b** Comparison of indel efficiencies of Cas12f1_v7 variants with one of the eleven mutations in a. **c** Indel frequencies of Cas12f1_v8 with sgRNA_ge4.1, sgRNA_S1a, sgRNA_S1b, sgRNA_S3, or sgRNA_design #2.

**d** Construction of enhanced Cas12f1 (eCas12f1). Data in the bar graphs represent the mean ± s.d. of three independent biological replicates. *P*-values were obtained using the two-tailed Student's t-test. ns, not significant. Source data are provided as a Source Data file.

enhanced AsCas12f variants: AsCas12f-YHAM (including F48Y, S188H, V232A, and E316M) and AsCas12f-HKRA (including I123H, D195K, D208R, and V232A)[22].

Referring to these two studies, we aimed to apply the mutations introduced in the three engineered AsCas12f variants (enAsCas12f, AsCas12f-YHAM, and AsCas12f-HKRA) to Cas12f1_v7. Through protein sequence alignment[23,24] of AsCas12f and Un1Cas12f1 (Supplementary Fig. 5), we identified eleven amino acids in the Un1Cas12f1 sequence corresponding to the mutations that enhanced the gene editing activity of AsCas12f, excluding one mutation (F48Y in AsCas12f-YHAM), as the corresponding amino acid is not aligned within the Un1Cas12f1 sequence. We then generated eleven Cas12f1_v7 variants by applying the same single mutations introduced in the enhanced AsCas12f to the corresponding amino acids in Cas12f1_v7 (Fig. 3a). We evaluated the gene editing efficiency of these eleven variants on three targets where Cas12f1_v7 exhibited the lowest, intermediate, and highest gene editing efficiencies (*RSPO4-1*, *sgVEGFA-1*, and *RMST*, respectively) among the ten targets tested in Fig. 2m. Experimental results showed that the variant with the L333A mutation introduced into Cas12f1_v7 exhibited a 2.44-fold increase in gene editing efficiency compared to Cas12f1_v7 in *RSPO4-1* (Fig. 3b). However, it showed a decrease in gene editing efficiency of 0.80-fold and 0.85-fold in *sgVEGFA-1* and *RMST*, respectively. In contrast, the variant with the S426G mutation introduced into Cas12f1_v7 also showed a 1.87-fold increase in gene editing efficiency in *RSPO4-1* compared to Cas12f1_v7, however no decrease in indel efficiency was observed in *sgVEGFA-1* and *RMST* (Fig. 3b). We further applied the mutations introduced to enhance the gene editing efficiency of OsCas12f1 (433 aa) from *Oscillibacter* sp. and RhCas12f1 (415 aa) from *Ruminiclostridium herbifermentans*[8] to Cas12f1_v7. We identified two mutations corresponding to UnCas12f1 from the protein alignment (Supplementary Fig. 5). We then constructed two Cas12f1_v7 variants (Supplementary Fig. 6a) and investigated the gene editing efficiency across three gene targets; however, the gene editing efficiency decreased for all targets (Supplementary Fig. 6b). Therefore, we named the variant with the S426G mutation introduced into Cas12f1_v7 as Cas12f1_v8.

Lastly, we tested five engineered sgRNAs with Cas12f1_v8 in three gene targets (Fig. 3c). We applied not only sgRNA_ge4.1[13] to Cas12f1_v8, but also sgRNA_S1a, sgRNA_S1b, and sgRNA_S3, which incorporate RNA stable hairpin sequences into sgRNA_ge4.1 (Fig. 1c–e). Additionally, we utilized the improved sgRNA_design #2 from CasMINI[12]. As a result, when sgRNA_S1b was applied to Cas12f1_v8, it showed the highest increase in gene editing efficiency in *RSPO4-1* and *RMST* compared to other sgRNAs (1.42-fold and 1.19-fold, respectively). A 1.11-fold increase was observed in *sgVEGFA-1*; however, this increase was not statistically significant (Fig. 3c).

Finally, we defined Cas12f1_v8 with sgRNA_S1b as enhanced Cas12f1 (eCas12f1) (Fig. 3d).

## Comparison of gene editing efficiency among SpCas9, AsCpf1 and Cas12f1 variants

To evaluate eCas12f1 rationally, we compared its gene editing efficiency with the widely utilized SpCas9, AsCpf1, and previously reported Cas12f1 variants (CasMINI_V3.1 and Cas12f_ge4.1). Given that these CRISPR systems recognize different PAM sequences (SpCas9 5′-NGG-3′; AsCpf1 5′-TTTN-3′; Cas12f1 variants 5′-TTTR-3′), we selected twenty-seven gene targets that allowed the CRISPR systems to share the same spacer sequence, of which twenty-two targets exhibited low indel efficiency for Cas12f1_ge4.1[13] (Fig. 4a and Supplementary Table 2). Additionally, we used the same plasmid copy number ($7.3 \times 10^{10}$ copies) for transfection to evaluate only the cleavage activity of each CRISPR system. Since the plasmid sizes of each CRISPR system differ (9172 bp for SpCas9, 8937 bp for AsCpf1, 6743 bp for CasMINI_V3.1, 6152 bp for Cas12f1_ge4.1, and 6685 bp for eCas12f1), the molecular weight corresponding to the same copy number also varies (685.8 ng for SpCas9, 668.0 ng for AsCpf1, 504.2 ng for CasMINI_V3.1, 460.0 ng for Cas12f1_ge4.1, and 500 ng for eCas12f1). Therefore, we used the transfection reagent in the same ratio for DNA amount of each CRISPR system. The experimental results showed that eCas12f1 exhibited significantly greater indel efficiency than CasMINI_V3.1 and Cas12f1_ge4.1, with average indel efficiencies of 4.14%, 25.34%, and 46.04% for CasMINI_V3.1, Cas12f1_ge4.1, and eCas12f1, respectively (Fig. 4b, c).

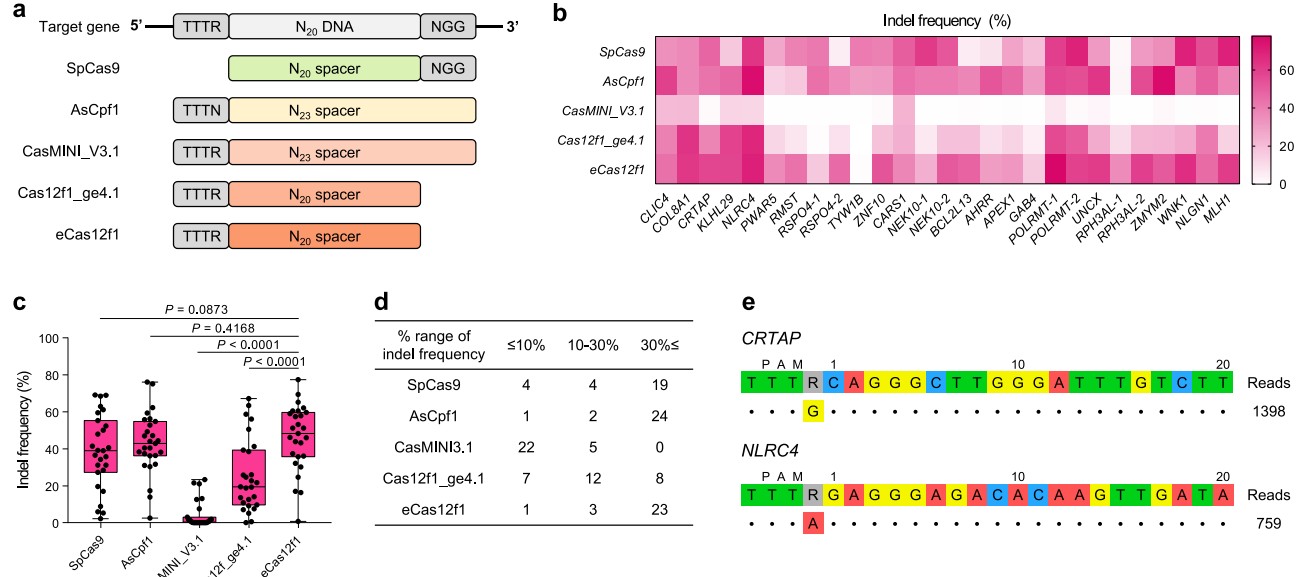

**Fig. 4 | Comparison of gene editing efficiency among SpCas9, AsCpf1, CasMINI, Cas12f1_ge4.1, and eCas12f1. a** Schematic illustration of the target gene shared by SpCas9, AsCpf1, and Cas12f1 variants. PAM sequences are shown in gray boxes. **b, c** Indel efficiency of SpCas9, AsCpf1, CasMINI_V3.1, Cas12f1_ge4.1, and eCas12f1 across twenty-seven gene targets. **d** Distribution of the number of target genes according to indel frequencies. **e** Off-target analysis for eCas12f1 targeting *CRTAP*

and *NLRC4*. The cleavage activity of eCas12f1 was detected only in on-targets. In the box-and-whisker plot, boxes represent the interquartile ranges with the median as a line, and whiskers extend from the minimum to the maximum values. *P*-values were obtained using the two-tailed Student's t-test. Source data are provided as a Source Data file.

We also confirmed that eCas12f1 exhibited gene editing activity comparable to SpCas9 and AsCpf1 at targets where the previously reported the two engineered Cas12f1 variants showed low gene editing efficiency (Fig. 4b, c). Additionally, eCas12f1 showed an editing efficiency of over 30% for more gene targets compared to the two Cas12f1 variants (Fig. 4d), confirming its effectiveness and utility as a gene editing tool.

To determine whether the small plasmid size of Cas12f1 provides an advantage in delivery into mammalian cells, we used three different plasmid delivery methods (lipid nanoparticle method, electro-physical method, and non-liposomal polymeric method) to compare the transfection efficiency of SpCas9, AsCpf1, Cas12f1_ge4.1, and eCas12f1. We transfected the same plasmid copy number of each CRISPR system into HEK293T cells. Since all CRISPR system plasmids encode the EGFP reporter, we counted the number of GFP-positive cells after 72 hr post-transfection and calculated the transfection efficiency. The results showed that all three delivery methods achieved high transfection efficiency. While eCas12f1 showed higher transfection efficiency than both SpCas9 and AsCpf1, a significant difference was only observed when compared to SpCas9 delivered using lipid nanoparticle and non-liposomal polymeric methods (Supplementary Fig. 7a–c). We also isolated GFP-positive cells from HEK293T cells treated with each CRISPR system using the three delivery methods and investigated the indel frequency at two gene targets (Supplementary Fig. 7d–f). The result demonstrated that, even with high intracellular DNA delivery efficiency, the gene editing efficiencies depend on the DNA cleavage capability of each CRISPR system.

The differences in gene editing efficiency among the four CRISPR systems delivered using the electro-physical and non-liposomal poly-meric methods were similar to those observed with the lipid nano-particle method employed throughout the development of eCas12f1. This suggests that even when using the electro-physical method and non-liposomal polymeric method for other gene targets, the gene editing efficiency of eCas12f1 would also be comparable to that of SpCas9 and AsCpf1, as indicated by the data in Fig. 4c. To summarize, eCas12f1 demonstrated higher transfection efficiency than SpCas9 and

comparable gene editing activity to both SpCas9 and AsCpf1, even with lower DNA amounts, indicating that it could serve as a cost-effective gene editing tool for future biotechnological applications and gene therapy.

We performed GUIDE-seq[25] to investigate off-target effects to assess the specificity and safety of eCas12f1. We selected two gene targets for which eCas12f1 exhibited a high gene editing efficiency (*CRTAP* and *NLRC4*). The GUIDE-seq analysis revealed that no off-target sites were detected with eCas12f1 (Fig. 4e), suggesting eCas12f1 offers specific and safe gene editing. This evidence supports the potential of eCas12f1 as a valuable tool in the field of genetic engineering.

## Application of eCas12f1 for cancer treatment

Polo-like kinase 1 (*PLK1*) belongs to a member of serine/threonine protein kinase family that are conserved in eukaryotic cells[21]. *PLK1* plays an important role in the cell cycle, and many studies have reported that inhibition of *PLK1* induces apoptosis of cancer cells by arresting in the mitotic phase. Inhibition of *PLK1* is also known as a candidate target for the treatment of breast cancer[26,27].

We evaluated the potential of eCas12f1 as a cancer therapy by targeting *PLK1* in breast cancer (SKBR-3) cell lines. To disrupt *PLK1* using eCas12f1, we designed sgRNA in exon3 of *PLK1* where cell cycle arrest and cell death in cancer cells were induced using SpCas9 in a previous study[28] (Fig. 5a). First, we confirmed the gene editing efficiency of eCas12f1-*PLK1* in HEK293T cells, and the results showed an indel efficiency of 34.62% (Supplementary Fig. 8a). Next, we transfected eCas12f1-*PLK1* into SKBR-3 cells through electroporation and analyzed the indel frequency of GFP-positive SKBR-3 cells after 72 hr post-transfection. As a result, *PLK1* knock out using eCas12f1 occurred at an indel frequency of 36.69% (Fig. 5b), and we confirmed the presence of small deletions (Supplementary Fig. 8b). The clonogenic assay results for SKBR-3 cells treated with eCas12f1-*PLK1* showed an 81.82% reduction in colony numbers compared to the non-targeting control (NTC), validating that eCas12f1 efficiently reduced the survival rate of cancer cells by interfering with the reproductive ability (Fig. 5c, d).

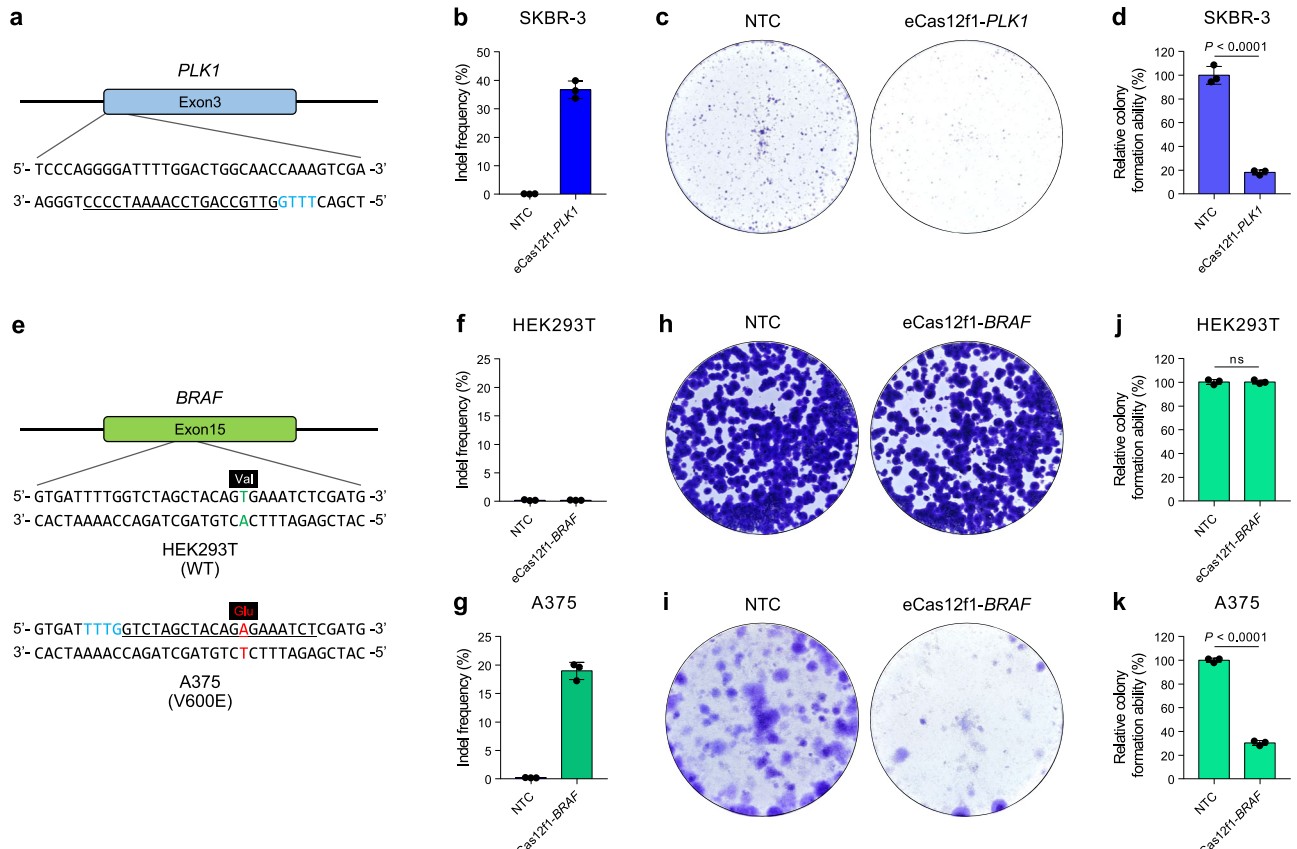

**Fig. 5 | Evaluation of eCas12f1 as a potential cancer therapy tool. a** Schematic illustration of sgRNA designed to disrupt exon3 of the *PLK1* gene. Underline, sgRNA spacer sequence; light blue letters, PAM sequence. **b** Indel frequency of eCas12f1-*PLK1* in SKBR-3 cells. 36.69% of indel frequencies were observed in GFP-positive SKBR-3 cells. NTC, non-targeting control. **c** Images of SKBR-3 colonies stained with crystal violet for the colony formation assay. **d** Quantification of the relative colony formation ability. **e** Schematic illustration of sgRNA designed to disrupt exon15 of the *BRAF* with V600E mutation. Underline, sgRNA spacer sequence; light blue letters, PAM sequence; light green letters, wild type base pair in *BRAF*; red letters, base pair mutation in *BRAF*; black box, amino acid code. **f, g** Indel frequency of eCas12f1-

*BRAF* in HEK293T cells (**f**) and A375 cells (**g**). 18.97% of indel frequencies were observed only in GFP-positive A375 cells. NTC, non-targeting control. **h, i** Images of HEK293T (**h**) and A375 (**i**) colonies stained with crystal violet for the colony formation assay after eCas12f1 treatment. **j, k** Quantification of the relative colony formation ability for HEK293T (**j**) and A375 (**k**) cells. 69.67% reduction in cell proliferation was observed only in A375 cells treated with eCas12f1 targeting *BRAF* V600E mutation gene. Data in the bar graphs represent mean ± s.d. of three independent biological replicates. *P*-values were obtained using the two-tailed Student's t-test. ns, not significant. Source data are provided as a Source Data file.

To verify whether *PLK1* knockout caused by eCas12f1 induced apoptosis in SKBR-3 cells, we stained the cells treated with either NTC or eCas12f1-*PLK1* using Annexin V, a calcium-dependent phospholipid-binding protein that binds to phosphatidylserine (PS) exposed on the outer membrane of apoptotic cells. We then counted the Annexin V-labeled cells using flow cytometry (Supplementary Fig. 8c). The results showed an 8.90% higher rate of apoptotic cells in the eCas12f1-*PLK1*-treated SKBR-3 cells compared to those treated with NTC, indicating that cell death was caused by eCas12f1-mediated *PLK1* knockout (Supplementary Fig. 8c, d).

The V600E mutation in B-raf proto-oncogene (*BRAF*) is the most common mutation in the *BRAF* gene and causes persistent activation and signal transduction, increasing cell proliferation and invasion in cancer patients[29]. The A375 cell line is a malignant melanoma derived from patient skin. This cell line has a single nucleotide mutation (c.7199 T > A) in exon15 of *BRAF*, resulting in a missense mutation that translates valine to aspartic acid (V600E mutation). In a previous study, it was reported that one of the widely used CRISPR systems, Cpf1, can specifically induce DNA cleavage in the genes with the *BRAF* V600E mutation[30]. To determine whether eCas12f1 effectively targets cancer-specific mutations and could be evaluated as a potential tool for cancer therapy, we designed an sgRNA targeting the *BRAF* V600E mutation in the A375 cell line (Fig. 5e and Supplementary Fig. 9a). A375 cells were

transfected with eCas12f1-*BRAF* by electroporation, and after 72 hr post-transfection, the indel efficiency of eCas12f1 at the *BRAF* gene was analyzed. HEK293T cells with the wild-type *BRAF* gene were used as a control (Supplementary Fig. 9b). The experimental results showed no gene editing activity was observed in HEK293T cells (Fig. 5f). However, in A375 cells, eCas12f1-*BRAF* showed a gene editing efficiency of 18.97%, indicating that eCas12f1 exhibited gene cleavage activity only in A375 cells with the *BRAF* mutation (Fig. 5g). Furthermore, the clonogenic assay results showed a 69.67% decrease in cell proliferation only in A375 cells treated with eCas12f1-*BRAF* (Fig. 5i, k), while no significant difference was observed in the HEK293T cells (Fig. 5h, j), demonstrating that eCas12f1 can be used effectively and safely in cancer therapy.

## Base editing using catalytically inactive eCas12f1

To determine whether eCas12f1 is compatible with efficient base editing, D326A and D510A mutations[9,31] were introduced into eCas12f1 to eliminate its gene cleavage activity. We then fused wild-type tRNA adenosine deaminase (TadA) and engineered TadA (TadA-8e)[32] to the N-terminus of eCas12f1 (eCas12f1-ABE) for adenosine base editing (Fig. 6a). The eCas12f1-ABE was verified on eight gene targets with a high adenine composition, selected from the gene targets used in the previous experiments. The overall average A-to-G conversion

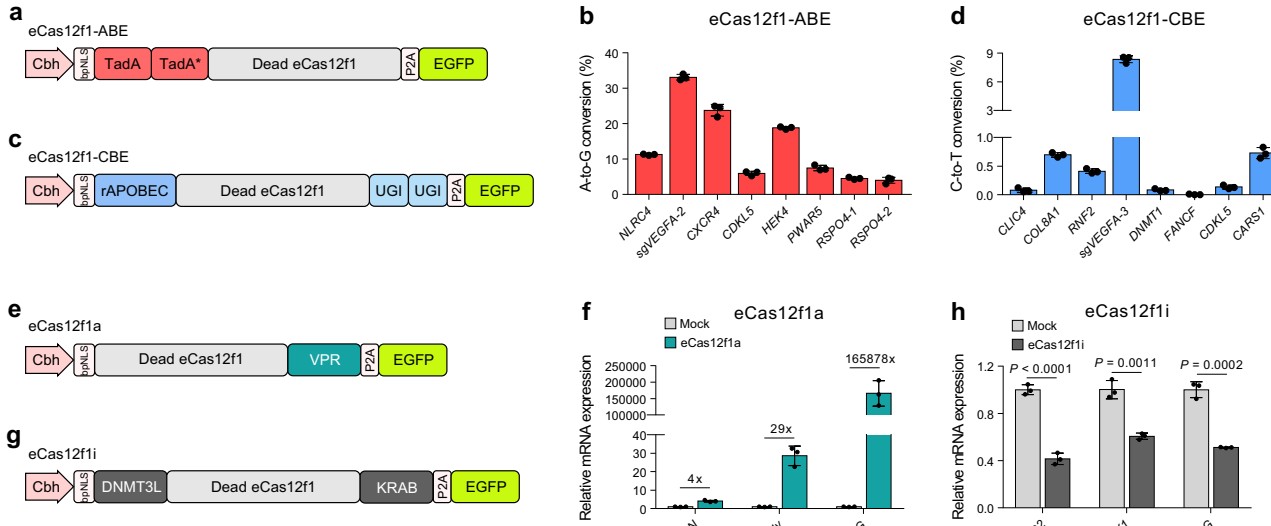

Fig. 6 | **Applications of eCas12f1 for base editing and regulation of gene expression. a** Constructs of adenine base editor with catalytically inactivated eCas12f1. TadA-TadA* was fused to the N-terminus of dead eCas12f1. TadA, wild type of TadA; TadA*, TadA-8e. **b** A-to-G conversion efficiency of eCas12f1-ABE at eight gene targets. **c** Construct of a cytosine base editor with catalytically inactivated eCas12f1. rAPOBEC and two copies of UGI were fused to the N- and C-termini of dead eCas12f1, respectively. **d** C-to-T conversion efficiency of eCas12f1-CBE in eight gene targets. **e** Construct of an eCas12f1 activator for activation of gene expression. VPR was fused to the C-terminus of dead eCas12f1. **f** Relative quantitation of mRNA level of three genes. The fold increase in gene expression is indicated above each bar graph. **g** Construct of an eCas12f1 inhibitor for interference with gene expression. DNMT3L and KRAB were fused to the N- and C-termini of dead eCas12f1, respectively. **h** Relative quantitation of mRNA level of three genes. Data represents mean ± s.d. of three independent biological replicates. *P*-values were obtained using the two-tailed Student's t-test. Source data are provided as a Source Data file.

efficiency was 13.64%, with a maximum efficiency of 33.13% observed in sg*VEGFA-2* (Fig. 6b). The A-to-G conversions by eCas12f1-ABE generated raging A3 to A14 with major conversions observed at positions A3 to A5 (Supplementary Fig. 10a, b)

Next, we fused rat apolipoprotein B mRNA editing enzyme 1 (rAPOBEC1) and two uracil DNA glycosylase inhibitors (UGI)[33] to N- and C-termini of eCas12f1, respectively (eCas12f1-CBE), for cytosine base editing (Fig. 6c).

We then evaluated eCas12f1-CBE in eight gene targets with a high composition of cytosine from the previously used gene targets. We confirmed a C-to-T conversion of up to 8.37% in sg*VEGFA-3*, while lower efficiencies of less than 1.00% in other targets, with no base conversion was observed in *FANCF* (Fig. 6d). The C-to-T conversions by eCas12f1-CBE occurred from C2 to C11, with major conversions observed at positions C3 to C5 (Supplementary Fig. 11a, b).

In conclusion, using inactivated eCas12f1 fused with deaminases, efficient adenine base conversion was observed. However, further research is required for effective cytosine base editing.

### Regulation of gene expression using catalytically inactive eCas12f1

Encouraged by effective base editing using dead eCas12f1, we evaluated whether efficient regulation of gene expression could be achieved using eCas12f1. We constructed an eCas12f1 activator (eCas12f1a) by fusing a tripartite activator, VP64-p65-Rta (VPR)[34], to the C-terminus of catalytically inactivated eCas12f1 (Fig. 6e) and verified the efficiency of eCas12f1a in three target genes. Effective gene activation was confirmed across all three target genes, and a maximum 165,878-fold increase in gene expression was observed in *HBG* compared to expression in the untreated cell pool (Fig. 6f).

We also constructed an eCas12f1 inhibitor (eCas12f1i) by fusing the DNA methyltransferase 3 like (DNMT3L) and Krüppel associated box (KRAB), both of which are known as transcriptional repressors[35], to the N- and C-termini of dead eCas12f1, respectively (Fig. 6g). We evaluated the repression of gene expression using eCas12f1i in three target genes, and the results demonstrated that gene expression was reduced by an

average of 48.85% in the three target genes compared to the untreated cell pool (Fig. 6h).

In conclusion, we identified that eCas12f1 can be used for efficient regulation of gene expression, as well as genome and base editing.

## Discussion

The CRISPR system is considered a highly innovative gene editing tool. It has recently gained worldwide attention, with the first clinical treatment approved in the United States. As the importance of the CRISPR system is emphasized, research on gene therapy is actively progressing with the aim of developing a more precise, safe, and stable CRISPR system.

Recently, many studies have reported CRISPR systems derived from various strains, and efforts have focused on discovering small CRISPR systems[7,8,31,36,37]. Compact CRISPR systems offer significant benefits for efficient delivery to mammalian cells, which addresses one of the major limitations of CRISPR technology. The effective loading of CRISPR components into delivery vehicles is limited by the large molecular sizes of CRISPR system[5]. Additionally, the production of nanoparticles for the larger mRNAs of the CRISPR system can result in increased impurities and a higher number of empty vesicles[6]. Furthermore, the CRISPR system is being used with functional factors, such as deaminases, reverse transcriptases, transcription factors, and fluorescence proteins for single-base conversions, diverse forms of gene editing, gene transcription regulation, and cell imaging. Therefore, there is an increasing need for compact CRISPR systems.

In this study, we aimed to increase the gene editing efficiency of compact Cas12f1, which consists of a 529 aa endonuclease, 161-nt tracrRNA, and 57-nt crRNA[9]. Initially, the wild-type Cas12f1 exhibited minimal gene editing activity in mammalian cells. Despite previous improvements reported in the Cas12f1 enzyme and sgRNA to boost its gene editing activity[12,13], challenges remained with low efficiency for certain target genes. In this study, we established four strategies to achieve high gene editing efficiency even for gene targets that previously showed low gene editing efficiency and applied them to an initial Cas12f1 variant consisting of codon optimized Cas12f1 and sgRNA_4.1.

We attempted to increase the structural stability of sgRNA by applying a highly stable hairpin to Stem 1 of Cas12f1 sgRNA, and the gene editing efficiency increased by an average of 1.45-fold relative to Cas12f1_v1. Introducing G181R mutation in Cas12f1_v1 to bolster interaction with the sgRNA spacer's 6th nucleotide resulted in an average of 2.43-fold efficiency improvement relative to Cas12f1_v1. Additionally, introduction of the D143R, T147R, and K330R mutations with G181R increased the gene editing efficiency by an average of 13.28-fold relative to Cas12f1_v1. Furthermore, we screened mutations that were introduced to enhance the gene editing activity of AsCas12f, a family member of Cas12f. We confirmed that the further introduction of S426G mutation into Cas12f1_v7 resulted in an additional increase in gene editing efficiency. Consequently, we defined the Cas12f1_v1 harboring the D143R, T147R, G181R, K330R, and S426G mutations with sgRNA_S1b, as eCas12f1. eCas12f1 exhibited significantly enhanced gene editing activity compared to CasMINI-V3.1 and Cas12f1_ge4.1. Additionally, it showed comparable editing efficiency to SpCas9 and AsCpf1, even with a smaller amount of DNA. This observation was particularly noted in gene targets where CasMINI-V3.1 and Cas12f1_ge4.1 exhibited low editing efficiency, thereby demonstrating the effectiveness of eCas12f1 as a robust genome editing tool with no off-target effect.

Finally, eCas12f1 was evaluated as a gene therapeutic, base editor, and regulator of gene expression. eCas12f1 induced cell death in breast cancer cells by disrupting *PLK1* and notably eCas12f1 effectively disrupted cancer-specific genes by recognizing *BRAF* mutations in the malignant melanoma cells. The dead eCas12f1 fused with deaminases exhibited a maximum adenine base conversion efficiency of 33.13%; however, further research is needed for efficient cytosine base editing. The dead eCas12f1 was also fused with transcriptional regulators and it performed effective up and down regulation of the target gene expression. In conclusion, eCas12f1 is a compact CRISPR system that performs gene editing comparable to widely used genome editors, even with a small amount of DNA. Its ability to utilize various effector proteins for effective base editing and gene regulations positions it as a powerful gene editing tool that could expand the tool kit for gene editing.

## Methods

### Plasmid construction

The DNA fragments encoding modified sgRNAs were produced by polymerase chain reaction (PCR) (Platinum™ SuperFi II PCR Master Mix, Invitrogen) using synthesized oligos (Bionics) and Cas12f-GE ver4.1 (Addgene, 176544) as a template. The human codon-optimized Cas12f1 sequence (GeneScript) was synthesized (Bionics) to construct Cas12f1_v1. The DNA fragments encoding Cas12f1 variants were produced using Platinum™ SuperFi II PCR Master Mix (Invitrogen, 12368050) and synthesized oligos (Bionics) and Cas12f1_v1 as a template. Using NEBuilder® HiFi DNA Assembly Master Mix (New England Biolabs, E2621L), plasmids encoding U6 promoter, modified sgRNA scaffolds, spacer cloning site with BsaI recognition sequences, chicken β-actin promoter, bipartite nuclear localization signal (bpNLS), Cas12f1 variants, 2xFLAG, P2A, and EGFP were cloned.

### Cell culture

HEK293T (ATCC, CRL-3216) and A375 (ATCC, CRL-1619) cells were cultured in Dulbecco's Modified Eagle Medium (Welgene, LM001-05) supplemented with 10% fetal bovine serum (Welgene, S101-01). SKBR-3 (ATCC, HTB-30) cells were cultured in McCoy's 5 A Medium (Welgene, LM005-01) supplemented with 10% fetal bovine serum. All the cell lines were maintained at 37 °C and 5% $CO_2$.

### Cell transfection

Lipofectamine™ 3000 Transfection Reagent (Invitrogen, L3000015) was used for HEK293T cells. $2 \times 10^4$ cells were seeded in a 24-well culture plate (Sarstedt, 83.3922) containing 500 μL of complete media. The day after the cell seeding, 500 ng plasmids was mixed with 50 μL Opti-MEM™ (Gibco, 31985070), 0.75 μL Lipofectamine 3000 reagent, and 1 μL P3000 reagent. The mixture incubated at RT for 15 min and treated to the HEK293T cells. The cells were then cultured for 72 hr. For NGS analysis, the cells were lysed using a lysis buffer (24 mM NaOH, 0.25 mM EDTA) and heated at 95 °C for 20 min. Then 50 mM HEPES were added to the cell lysates for NGS.

Electroporation was performed using Neon™ Transfection System (Thermo Fisher Scientific, MPK1096). $2 \times 10^5$ cells and 500 ng plasmid were used for 10 μL reaction. Electroporation conditions were 1150 V, 10 mA, 3 pulses for HEK293T, 1050 V, 30 mA, 3 pulses for SKBR-3, and 1200 V, 20 mA, 3 pulses for A375 cells. The cells then were seeded in 12-well culture plate (Sarstedt, 83.3921) and cultured for 72 hr.

We also used TransIT-X2 Dynamic Delivery System (Mirus, MIR 6000) for non-liposomal polymeric DNA delivery method according to the manufacturer's instruction. One day before transfection, $5 \times 10^4$ HEK293T cells were seeded in 12-well culture plate (Sarstedt, 83.3921). 500 ng plasmid was mixed with 1.5 μL TransIT-X2 reagent and 50 μL Opti-MEM™ (Gibco, 31985070). The mixture incubated at RT for 30 min and treated to the HEK293T cells. The cells were then cultured for 72 hr.

### Flow cytometric analysis

We analyzed the transfection efficiency of three different delivery methods. Plasmids of SpCas9, AsCpf1, Cas12f1_ge4.1, and eCas12f1 encoding EGFP were transfected into HEK293T cells using Lipofectamine™ 3000, Neon™ Transfection System, and TransIT-X2 Dynamic Delivery System. After 72 hr, GFP-positive cells were counted from 10,000 events using BD FACSCanto™ II (BD Biosciences) and analyzed by FlowJo™ v10.10.0 software.

### High-throughput sequencing

Cell lysates containing genomic DNA were used to amplify genes of interest using KAPA HiFi HotStart PCR Kit (Roche, KR0369). The genomic regions of interest were amplified using three-round PCR strategy to add Illumina adapters and unique sample barcodes. The sequences of sample libraries were analyzed by MiSeq system (Illumina). Gene editing efficiency was analyzed using EUN. The primers used for sequencing are shown in Supplementary Data 1.

### GUIDE-seq

GUIDE-seq was conducted following previously reported protocols[25]. 500 ng of plasmid encoding each Cas nuclease and sgRNA was transfected with 5 pmol of 5′ end-protected dsODN into $2 \times 10^5$ HEK293T cells using Neon™ Transfection System (Thermo Fisher Scientific). Cells were harvested 72 hr post-transfection and genomic DNA (gDNA) was isolated using DNeasy Blood & Tissue Kit (Qiagen, 69506). Targeted deep sequencing was conducted to estimate indel and dsODN integration frequencies. 2 μg of gDNA in 650 μL TE buffer was fragmented using ultrasonic homogenizer (KBT, KUS-650) with Φ 2 tip (100 W, on for 1 sec, off for 4 sec, treat for 25 min). After purification of sheared DNA using Mag-Bind® Total Pure NGS (Omega Bio-Tek, M1378-01), end-repair and A-tailing were performed using Fast DNA End Repair Kit (Thermo Scientific, K0771) and Y-adapters were ligated using T4 DNA Ligase (New England Biolabs, M0202L). Two rounds of nested anchored PCR were performed for target enrichment of plus and minus strands separately using Platinum™ Taq DNA Polymerase (Invitrogen, 10966026). The libraries were prepared by pooling an equal number of molecules of the plus and minus reactions, calculated using KAPA Library Quantification Kit Illumina® platforms (Kapa Biosystems, KK4824), and sequenced for paired-end 150-cycle on an Illumina MiSeq system. Data were analyzed and visualized using open-source GUIDE-seq software[38].

## Colony formation assay

One day after transfecting HEK293T and SKBR-3 cells with vectors encoding eCas12f1-*PLK1* or non-targeting eCas12f1, the cells were trypsinized and washed with phosphate-buffered saline (PBS). Using BD FACSAria™ III (BD Biosciences), $5 \times 10^3$ GFP-positive cells were sorted and seeded in 6-well culture plates (Sarstedt, 83.3920) and cultured at 37 °C and 5% $CO_2$. After 14 days, the cells were washed twice with PBS and fixed with 4% paraformaldehyde for 10 min. Then the cells were stained with 0.5% crystal violet for 10 min. After washing the cells twice with PBS, the areas of colonies were calculated using ImageJ software.

For the HEK293T and A375 cells treated with eCas12f1-*BRAF*, assays were performed 10 days after GFP-positive cell seeding with following the same methodology above.

## Cellular apoptosis assay

The dead Cell Apoptosis Kits with Annexin V (Invitrogen, V13242) were used to assess cellular apoptosis. After 48 hr post-transfection using Neon™ Transfection System (Thermo Fisher Scientific), the cells were harvested and washed with cold PBS. Then the cells were centrifuged for 2 min at $300 \times g$ and the supernatant was removed. The cell pellet was resuspended with 100 µL of 1X annexin-binding buffer. Subsequently, 5 µL of Alexa Fluor® 488 annexin V and 1 µL of the 100 µg/mL Propidium iodide (PI) were added to the cells. The cells were incubated at room temperature for 15 min. Following incubation, 400 µL of 1X annexin-binding buffer was added, and the samples were mixed gently. The stained cells were kept on ice during analysis and annexin-labeled cells were counted (10,000 events) using BD FACSCanto™ II (BD Biosciences).

## Quantitative real-time PCR

Total RNA was extracted from transfected HEK293T cells using TaKaRa MiniBEST Universal RNA Extraction Kit (TaKaRa, 9767 A). The extracted total RNA was then used to synthesize cDNA using the iScript™ cDNA Synthesis Kit (Bio-Rad, 1708890). qPCR reaction mixtures were prepared using AccuPower® 2X Greenstar™ (Bioneer, K-6254). The qPCR was performed using a CFX Connect Real-Time PCR Detection System (BioRad) and protocol included preheating at 95 °C for 3 min, followed by 40 cycles at 95 °C for 10 s and 60 °C for 30 s. For transcripts with low expression levels, quantification cycle values exceeding 35 were considered as 35.

## Statistics and reproducibility

All experiments were performed with at least three independent biological replicates, unless otherwise noted. Statistical significance was analyzed using two-tailed Student's t-test. A *P* value of < 0.05 was considered statistically significant. The bar and dot plots were presented as the mean ± standard deviation (s.d.). The box and dot plots were presented as interquartile ranges with the median indicated by a line and whiskers extending from the minimum to the maximum values. No statistical method was used to predetermine sample size. No data were excluded from the analyzes.

## Reporting summary

Further information on research design is available in the Nature Portfolio Reporting Summary linked to this article.

## Data availability

The NGS data and GUIDE-seq generated in this study have been deposited in the NCBI Sequence Read Archive (SRA) under accession number PRJNA1104538. All data supporting the findings of this study are provided within the paper, Supplementary Information, and Supplementary Data. Source data are provided with this paper.

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

## Acknowledgements

This study was supported by the Chung Yang, Cha Young Sun & Jang Hi Joo Memorial Fund (K.K.) and the National Research Foundation of Korea (NRF) grants funded by the Korea government (MSIT) (NRF-2023R1A 2C2004222 (K.K.), RS-2023-00220894 (K.K. and H.S.K), RS-2023-00261905 (K.K.), RS-2023-NR077033 (K.K.), and RS-2024-00441068 (K.K.)). The illustrations in Fig. 1a were created with BioRender.com.

## Author contributions

S.-J.P. and K.K. designed the study. S.-J.P., S.J., W.J.J., T.Y.J., J.H.L., and J.Y. performed the experiments. K.K. supervised the research. S.-J.P., S.J., D.E.Y., H.L., J.C., H.S.K., and K.K. discussed the results and commented on the manuscript.

## Competing interests

S.-J.P. and K.K. have submitted a patent application to the Korea patent office pertaining to the enhanced Cas12f1 variants and their uses described in this work (application number 10-2024-0058649). The remaining authors declare no competing interests.
