## [Transparent Peer Review file · Nature Communications]

Robust genome editing activity and the applications of enhanced miniature CRISPR-Cas12f1

Corresponding Author: Professor Kyoungmi Kim

Version 0:

Reviewer comments:

Reviewer #1

(Remarks to the Author)

Kim et al. engineer Cas12f1 gRNAs to be more efficient and describe a Cas12f1 variant with improved editing efficiency that is also compatible with base editing and transcriptional modulation. The study presents a robust amount of data and the figures are clear. As the authors state, improved Cas12f variants are potential very useful for therapeutic genome editing due to their small size. However, there are already many published improved versions of Cas12f variants and related improved gRNAs which could limit the impact of the study. One way to increase relevance would be to try to combine not just some, as done here already, but all published ways to potentially improve Cas12f protein or gRNA efficiency.

Major points

1. I miss a clear and comprehensive overview of the state-of-the-art of improved Cas12f variants. Also, state-of-the-art efficiencies of Cas12f editing seem to be belittled. Reference 13 of the study already presented efficiencies comparable to that of SpCas9, but the introduction of this manuscript describes achieved efficiencies prior to this study as 'minimal' or 'marginal'. There are several studies that achieve good efficiencies with improved Cas12f variants and gRNAs: e.g. PMID: 37776859, PMID: 37400536, PMID: 34475560. It would be great if the authors would try to experimentally combine not just some, as done here already, but all published ways to potentially improve Cas12f protein or gRNA efficiency (described mutations in other Cas12f species might be mappable onto the Cas12f1 structure).
2. The authors should also try to incorporate the published super-stable hairpin (5'-GGAC(UUCG)GUCC-3') they refer to in stem 1. It is unclear why they choose to do the gRNA base substitutions as done in S1a or S1b. The authors should provide a rationale for this. I can follow to exchange a A:U with a G:C base pair (more hydrogen bonds), but this was only one of the changes done.
3. It seems far-fetched to present PLK editing in HEK293T cell as potential way for cancer treatment with Cas12f1 - it is just one more target the authors show robust editing for. PLK targeting would result in knockout of PLK in cancer, but also healthy wild type cells. Such a cancer treatment claim could be experimentally shown by packing Cas12f1 into LNPs coated with specific cell-targeting antibodies. A much easier way would be to target a cancer mutation, so that the wild type sequence is not cut by the CRISPR complex.

Minor points:

-The authors write that, Cas12f1_ge4.1 was developed by optimizing the sgRNA of Cas12f1 and increased indel activity by up to 867-fold', while the original study they refer to states that 'These optimizations synergistically increased the average (!) indel frequency by 867-fold'.

-line 351: typo CPRSPR

Reviewer #2

(Remarks to the Author)

Park et al. describe the development of a miniature Cas12f1 variant with enhanced genome editing activity in mammalian cells. Cas12f1 was previously identified as a small CRISPR RNA-guided nuclease with limited genome editing efficiency in mammalian cells. In the past years several publications have described attempts to improve the gene editing efficiency of Cas12f1, with limited success. Park et al. demonstrate the structured and incremental enhancement of the genome editing efficiency of Cas12f1 by modifying the structure of the sgRNA and introducing mutations in critical domains of the Cas12f1 coding sequence. In doing so, they eventually design a small Cas12f1 variant (eCas12f1) with genome editing efficiency comparable to that of conventional SpCas9 and AsCpf1, that is also suitable for use in base editing and gene activation/repression systems.

The development of this efficient miniature Cas12f1 variant represents a significant and relevant improvement over currently existing systems, and adds a valuable tool to the ever-expanding CRISPR toolbox, with potential future uses ranging from biotechnological applications to gene therapy.

Overall, the manuscript is well written and structured, and conveys a clear message. I have several suggestions to further improve the manuscript prior to publication.

Major issues:

- The main advantage of the miniature eCas12f1 over other, larger, CRISPR RNA-guided nucleases is the supposed higher delivery efficiency to mammalian cells. The authors show that their eCas12f1 is smaller than SpCas9 and AsCpf1, while possessing similar genome editing efficiency, but do not actually demonstrate improved delivery efficiency. The authors should directly compare delivery efficiency of eCas12f1 to -at least- SpCas9, AsCpf1 and a previously developed Cas12f1 variant, preferably using multiple delivery methods, to actually demonstrate the superiority of eCas12f1 for future biotechnological and (pre)clinical applications.
- The authors demonstrate indel generation efficiencies of 13.76% and 42.53% in PLK1 in U87 and SKBR-3 cells, respectively (figure 5e-h), but do not show the transfection efficiency. Based on figures 5c and 5d, the transfection efficiency seems rather low, especially in U87 cells. As such, the authors may underestimate the gene editing efficiency of eCas12f1. Ideally, successfully transfected cells are sorted to adequately assess gene editing efficiency.

Minor issues:

- It is not always clear why certain genes are chosen to evaluate the editing efficiency of Cas12f1 variants. For example, gene editing efficiency of codon-optimized Cas12f1 and later Cas12f1_ge4.1 and Cas12f1_v1 are tested against a slightly different panel of genes (Suppl. Fig S1b,c) than Cas12f1_v1, Cas12f1_S1a and Cas12f1_S1b (Fig. 1b). The authors should state how and why a certain panel of target genes was chosen.
- PLK1 is a well known therapeutic target in breast cancer and glioblastoma cells, whose knockout almost universally leads to cell cycle arrest and/or cell death. The authors use colony-formation assays to demonstrate reduced survival rate of U87 and SKBR-3 cells upon eCas12f1-induced PLK1 knockout. Colony-forming assays, however, do not discriminate well between cell death and cell cycle arrest. The authors should either address this textually, or perform experiments to demonstrate cell death, apoptosis and/or cell cycle arrest upon PLK1 knockout. Ideally, sorted, successfully transfected cells are used for such experiments.
- Knockout of PLK1 in U87 and SKBR-3 cells may lead to cell death in less than 72 hours. As such, the authors may underestimate the indel generation efficiency of eCas12f1, because negative selection of successfully edited cells may have occurred (Fig. 5g,h). This would also explain the rather large effect of PLK1 knockout on colony-forming potential of U87 cells, at an apparently low indel generation efficiency. It would be interesting to evaluate indel generation efficiency or PLK1 expression at an earlier timepoint.
- CRISPR is misspelled in line 351.

Reviewer #3

(Remarks to the Author)

Reviewer #4

(Remarks to the Author)

Compact CRISPR-Cas tools have great gene therapy potential due to their high-efficiency delivery capacity. UnCas12f1 is a previously discovered mini Cas enzyme. In this manuscript, the authors Park et al. optimized UnCas12f1 by improving gRNA and target DNA binding affinities in combination with previously identified mutations. The eCas12f1 shows high editing efficiency and is compatible with base editing and gene activation, which could be useful for the gene editing field. Before recommending it for publication, I have a few concerns.

1. Off-target detection: the use of Cas-OFFinder with target sequencing may not provide comprehensive off-target identification. The authors may consider using de novo off-target identification methods such as GUIDE-seq, PEM-seq, CIRCLE-seq, or Digenome-seq for a more thorough off-target activity profile of eCas12f1.

2. When comparing the editing efficiency of various CRISPR-Cas, the authors need to normalize the transfection rates. Cas9 and Cpf1 plasmids are larger in size compared to Cas12f1 tools, which may lead to lower levels of transfection.

3. minor point: it's confusing to read the one-color figure panel in Fig. 2g, 2h, or similar ones. It's better to use a two-color system: >1 red, <1 blue.

Version 1:

Reviewer comments:

Reviewer #1

(Remarks to the Author)

The authors have addressed all of my comments.

Reviewer #2

(Remarks to the Author)

The authors have carefully considered all of my comments and have performed a substantial amount of additional experiments to address my concerns. Although the transfection efficiency of eCas12f1 is only marginally superior to spCas9 and AsCpf1, the resulting gene editing efficiency of at least one target is clearly higher. Additionally, HEK293T cells are relatively easy to transfect, and as such, the enhanced delivery efficiency may be more marked in cells that are difficult to transfect. In the end it is up to the research community to determine which system works best for each individual research question, and eCas12f1 represents a promising addition to the CRISPR toolbox.

The additional experiments concerning the effect of PLK1 KO using the eCas12f1 system on the clonogenic capacity of SKBR-3 cells are convincing and add to the strength of the manuscript.

Finally, the authors have adequately addressed my minor concerns, either by explaining their choices textually or performing additional experiments.

In conclusion, I would like to congratulate the authors on their work, and I consider the revised manuscript suitable for publication.”

Reviewer #3

(Remarks to the Author)

Reviewer #4

(Remarks to the Author)

The authors have made great efforts to address the concerns and I think the study is much improved. I have no more comments and recommend the publication of it in Nature Communications.

Point-by-point Responses

We would like to sincerely thank the Reviewers for their valuable comments and suggestions, which have greatly improved the quality of our study. We have carefully and thoroughly addressed all the concerns in the revised manuscript, as well as in the responses below.

Reviewer comments

Reviewer #1

Kim et al. engineer Cas12f1 gRNAs to be more efficient and describe a Cas12f1 variant with improved editing efficiency that is also compatible with base editing and transcriptional modulation. The study presents a robust amount of data and the figures are clear. As the authors state, improved Cas12f variants are potential very useful for therapeutic genome editing due to their small size. However, there are already many published improved versions of Cas12f variants and related improved gRNAs which could limit the impact of the study. One way to increase relevance would be to try to combine not just some, as done here already, but all published ways to potentially improve Cas12f protein or gRNA efficiency.

Major points 1:

I miss a clear and comprehensive overview of the state-of-the-art of improved Cas12f variants. Also, state-of-the-art efficiencies of Cas12f editing seem to be belittled. Reference 13 of the study already presented efficiencies comparable to that of SpCas9, but the introduction of this manuscript describes achieved efficiencies prior to this study as 'minimal' or 'marginal'.

Response:

We appreciate your comments regarding the innovative aspects of our research and your suggestions on how to strengthen them further. The gene editing efficiency of the 'unmodified' natural Un1Cas12f1 in mammalian cells is 'marginal' (line 71) or 'minimal'(line 476) (nearly 0%). As you mentioned and as we explained in our introduction (line 72-80), two studies (reference 12 and 13) have already reported improved Un1Cas12f1 variants exhibiting increased gene editing efficiency in mammalian cells. However, depending on the target gene, low editing activity was still

observed. For example, Cas12f1_ge4.1 from reference 13 (PMID: 34475560), which you mentioned, exhibited below 10% indel efficiency at 19 out of 88 gene targets and CasMINI_V3.1 from reference 12 exhibited below 10% indel efficiency at all 8 gene targets (Thus, we only changed 'minimal' in the description of Cas12f1_ge4.1 to 'low' (line 76)). Therefore, the purpose of our study is to address this issue and enhance the efficient application of Cas12f1 as a gene therapy tool, thereby expanding its potential across a wider range of gene targets.

To this end, we focused on gene targets where the engineered Cas12f1_ge4.1 from reference 13 (PMID: 34475560) showed low editing efficiency and conducted experiments to engineer the Cas12f1 protein and sgRNA. During this process, additional amino acid mutations were introduced in a stepwise manner to the initial version, Cas12f1_v1. For a clear and comprehensive overview of the state-of-the-art of improved Cas12f variants in this study, we assigned higher numbers to the position of # in the format Cas12f1_v# to clarify the changes made at each stage. For example, we named Cas12f1_v3, which is Cas12f1_v1 with three amino acid mutations in the DNA binding pocket, and Cas12f1_v7, which is Cas12f1_v3 with G181R mutation to enhance affinity with the sgRNA spacer (Fig. 2).

Major points 1:

There are several studies that achieve good efficiencies with improved Cas12f variants and gRNAs: e.g. PMID: 37776859, PMID: 37400536, PMID: 34475560. It would be great if the authors would try to experimentally combine not just some, as done here already, but all published ways to potentially improve Cas12f protein or gRNA efficiency (described mutations in other Cas12f species might be mappable onto the Cas12f1 structure).

Response:

We also appreciate your valuable suggestion to consider other studies aimed at improving the gene editing efficiency of various Cas12f species. By incorporating insights from the references, you mentioned (PMID: 37776859, PMID: 37400536), we introduced mutations previously applied to enhance the activity of Acidibacillus sulfuroxidans Cas12f (AsCas12f), another member of the Cas12f family, into Cas12f1_v7 step by step (Fig. 3a,b) (line 281-298). This approach led us to identify a variant containing the S426G mutation in Cas12f1_v7, which exhibited further improved

gene editing efficiency, and we named this variant Cas12f1_v8 (Fig. 3b).

We further applied the mutations introduced to enhance the gene editing efficiency of OsCas12f1 from *Oscillibacter* sp. and RhCas12f1 from *Ruminiclostridium herbifermentans* (reference 9) to Cas12f1_v7. We then constructed two Cas12f1_v7 variants (Extended Data Fig. 6a) and investigated the gene editing efficiency across three gene targets; however, the gene editing efficiency decreased for all targets (Extended Data Fig. 6b) (line 298-305).

Finally, we combined Cas12f1_v8 with five engineered sgRNAs (sgRNA_ge4.1 from reference 12 (PMID: 34475560), sgRNA_design #2 from reference 13 and three new sgRNAs designed in this study to enhance the structural stability of sgRNA) (Fig. 3c). Experimental results showed that the combination of Cas12f1_v8 and sgRNA_S1b yielded the highest gene editing efficiency, leading us to name this enhanced variant "eCas12f1" (Fig. 3c,d) (line 306-315).

Thanks to your input, we were able to develop eCas12f1, which exhibits gene editing efficiency comparable to that of SpCas9 and AsCpf1, even for gene targets where previously reported improved Cas12f1 showed low efficiency (Fig. 4a-d) (line 318-338).

Major points 2:

The authors should also try to incorporate the published super-stable hairpin (5'-GGAC(UUCG)GUCC-3') they refer to in stem 1. It is unclear why they choose to do the gRNA base substitutions as done in S1a or S1b. The authors should provide a rationale for this. I can follow to exchange a A:U with a G:C base pair (more hydrogen bonds), but this was only one of the changes done.

Response:

Thank you for pointing out the important aspect that we overlooked. The 5'-GGAC(UUCG)GUCC-3' sequence is a hairpin structure that is exceptionally stable and a very common RNA hairpin structure (reference 18). In the study we referenced (reference 17) for introducing super stable RNA hairpins, we observed that modifying part of the sequence of 5'-GGAC(UUCG)GUCC-3', even C:G to G:C base pair, resulted in a subtle decrease in the gene cleavage activity of SpCas9. This led us to speculate that modifications in the stable hairpin sequence reduce the stability of the hairpin

structure. Therefore, considering the scaffold structure of the Cas12f1 sgRNA, we partially incorporated the original sequence of the super stable RNA hairpin into the Stem 1 and stem3 of the Cas12f1 sgRNA (Fig. 1d,e). First, we designed two stems for Stem 1. One approach was to retain the loop sequence without modification, as A⁻⁸⁶ in the loop interacts with K330 of Cas12f1.2; thus, only the stem sequence of the stable hairpin was introduced by replacing U⁻⁹⁴:A⁻⁸² and C⁻⁹³:G⁻⁸³ with G:C base pairs (Cas12f1_S1a). The other approach involved an additional change in the loop sequence by substituting A⁻⁸⁸ with U to incorporate the UUCG loop sequence of the stable hairpin (Cas12f1_S1b). As a result, Cas12f1_S1b exhibited higher gene editing activity than Cas12f1_S1a (Fig. 1f,g). The Stem 1 sequences did not change significantly, as you have pointed out; however, we found that the exceptional structural stability characteristics conferred by the super stable hairpin sequence itself resulted in increased sgRNA stability. This likely explains why Cas12f1_S1b, which has the same sequence as the stable hairpin except for A⁻⁸⁶ in the loop of Stem 1, exhibits higher gene editing efficiency than Cas12f1_S1a. This information also has been added in line 137-158.

Major points 3:

It seems far-fetched to present *PLK* editing in HEK293T cell as potential way for cancer treatment with Cas12f1 - it is just one more target the authors show robust editing for. *PLK* targeting would result in knockout of *PLK* in cancer, but also healthy wild type cells. Such a cancer treatment claim could be experimentally shown by packing Cas12f1 into LNPs coated with specific cell-targeting antibodies. A much easier way would be to target a cancer mutation, so that the wild type sequence is not cut by the CRISPR complex.

Response:

Thank you for your comments that provide an opportunity to demonstrate the potential and excellence of the eCas12f1 as a prospective cancer treatment. As you mentioned, targeting *PLK1* is not suitable as a genetic target for cancer therapy due to its activity in normal cells, which raises safety concerns. Therefore, we sought to identify cancer cell-specific mutations. We referenced a previous study that successfully targeted and disrupted the *BRAF* V600E mutation specifically in the malignant melanoma A375 cell line, using the widely used CRISPR system, Cpf1 (reference 32). To evaluate the cancer

cell-specific gene editing activity of eCas12f1, we designed sgRNA targeting the *BRAF* V600E mutation in A375 cell line (Fig. 5e). A375 cells were transfected with eCas12f1-*BRAF* via electroporation, and 72h post-transfection, the indel efficiency of eCas12f1 at the *BRAF* gene was analyzed. HEK293T cells were tested together as a control with the wild-type *BRAF* gene (Extended Data Fig. 9b). The experimental results showed that no gene editing activity was observed in HEK293T cells, which do not have mutations in the *BRAF* gene (Fig. 5f). However, in A375 cells, eCas12f1-*BRAF* showed a gene editing efficiency of 18.97%, indicating that eCas12f1 exhibited gene cleavage activity only in A375 cells with *BRAF* mutations (Fig. 5g). Furthermore, the results of the clonogenic assay showed that cell proliferation ability decreased by 30.33% only in A375 cells treated with eCas12f1-*BRAF*, indicating that eCas12f1 can be used effectively and safely in cancer therapy (Fig. 5h-k). This information also has been added in line 397-416.

Minor points 1:

The authors write that , Cas12f1_ge4.1 was developed by optimizing the sgRNA of Cas12f1 and increased indel activity by up to 867-fold', while the original study they refer to states that 'These optimizations synergistically increased the average (!) indel frequency by 867-fold'.

Response:

As you mentioned, we have corrected the phrase from "increased indel activity by up to 867-fold" to "increased the average indel frequency by 867-fold" in line 100. Thank you for pointing out this mistake.

Minor points 2:

line 351: typo CPRSPR

Response:

As you mentioned, we have corrected "CPRSPR" to "CRISPR" in line 464. Thank you for pointing out this typo.

Reviewer #2

Park et al. describe the development of a miniature Cas12f1 variant with enhanced genome editing activity in mammalian cells. Cas12f1 was previously identified as a small CRISPR RNA-guided nuclease with limited genome editing efficiency in mammalian cells. In the past years several publications have described attempts to improve the gene editing efficiency of Cas12f1, with limited success. Park et al. demonstrate the structured and incremental enhancement of the genome editing efficiency of Cas12f1 by modifying the structure of the sgRNA and introducing mutations in critical domains of the Cas12f1 coding sequence. In doing so, they eventually design a small Cas12f1 variant (eCas12f1) with genome editing efficiency comparable to that of conventional SpCas9 and AsCpf1, that is also suitable for use in base editing and gene activation/repression systems.

The development of this efficient miniature Cas12f1 variant represents a significant and relevant improvement over currently existing systems, and adds a valuable tool to the ever-expanding CRISPR toolbox, with potential future uses ranging from biotechnological applications to gene therapy.

Overall, the manuscript is well written and structured, and conveys a clear message. I have several suggestions to further improve the manuscript prior to publication.

Major points 1:

The main advantage of the miniature eCas12f1 over other, larger, CRISPR RNA-guided nucleases is the supposed higher delivery efficiency to mammalian cells. The authors show that their eCas12f1 is smaller than SpCas9 and AsCpf1, while possessing similar genome editing efficiency, but do not actually demonstrate improved delivery efficiency. The authors should directly compare delivery efficiency of eCas12f1 to -at least- SpCas9, AsCpf1 and a previously developed Cas12f1 variant, preferably using multiple delivery methods, to actually demonstrate the superiority of eCas12f1 for future biotechnological and (pre)clinical applications.

Response:

Thank you for suggesting ways to further demonstrate the superiority of eCas12f1. Since plasmids were used to compare the gene editing activities of various CRISPR systems in our study, we aimed to investigate whether there were differences in

transfection efficiency according to the DNA size of each CRISPR system (Total vector sizes: SpCas9 is 9172 bp, AsCpf1 is 8934 bp, Cas12f1_ge4.1 is 6152 bp, and eCas12f1 is 6685 bp) by comparing three different transfection methods (Lipid nanoparticle method: Lipofectamine 3000, Electro-physical method: Neon electroporation, Non-liposomal polymeric method: TransIT-X2). To ensure a fair comparison, we used the same copy number of DNA for each CRISPR system and adjusted the volumes of Lipofectamine 3000 and TransIT-X2 reagent (we used 1 μ l of P3000 reagent per 500 ng plasmid for Lipofectamine3000 and 1.5 μ l of TransIT-X2 reagents per 500 ng plasmid for TransIT-X2) according to the manufacturer's protocols, excluding Neon electroporation, since the buffer volume could not be adjusted. We then harvested the HEK293T cells after 72 hr post-transfection and counted the number of GFP+ cells to assess the transfection efficiency. The results showed that all three delivery methods achieved high transfection efficiency. While eCas12f1 appeared to have higher transfection efficiency than both SpCas9 and AsCpf1, a significant difference was only observed when compared to SpCas9 with lipid nanoparticle and non-liposomal polymeric methods (Extended Data Fig. 7a-c). We also isolated GFP+ cells from HEK293T cells treated with each CRISPR system using the three delivery methods and investigated the indel frequency at two gene targets (Extended Data Fig. 7d-f). The result demonstrated that, even with high intracellular DNA delivery efficiency, the gene editing efficiencies depend on the DNA cleavage capability of each CRISPR system. The differences in gene editing efficiency among the four CRISPR systems delivered using the electro-physical method and non-liposomal polymeric method were similar to those observed with the lipid nanoparticle method, employed in the entire process for the development of eCas12f1. This suggests that even when using the electro-physical method and non-liposomal polymeric method for other gene targets, the gene editing efficiency of eCas12f1 would also be comparable to that of SpCas9 and AsCpf1, as indicated by the data in Fig. 4c. In summary, eCas12f1 demonstrated higher transfection efficiency to SpCas9 and comparable gene editing activity to SpCas9 and AsCpf1, even with a lower amount of DNA, indicating that it could serve as a cost-effective gene editing tool for future biotechnological applications and gene therapy. This information also has been added in line 339-363.

Once again, we appreciate your valuable suggestions, which have helped us further strengthen our study and clarify the advantages of eCas12f1 for future applications.

Major points 2:

The authors demonstrate indel generation efficiencies of 13.76% and 42.53% in *PLK1* in U87 and SKBR-3 cells, respectively (figure 5e-h), but do not show the transfection efficiency. Based on figures 5c and 5d, the transfection efficiency seems rather low, especially in U87 cells. As such, the authors may underestimate the gene editing efficiency of eCas12f1. Ideally, successfully transfected cells are sorted to adequately assess gene editing efficiency.

Response:

We appreciate your suggestion of appropriate evaluation methods for eCas12f1. Following your comments, we aimed to assess the gene editing efficiency of eCas12f1 using only successfully transfected cells. We transfected eCas12f1 and sgRNA-*PLK1* via electroporation and sorted GFP+ cells after 24 hr. However, in the case of U87, the cell damage caused by electroporation was significant, resulting in an insufficient number of GFP+ cells. Consequently, we were only able to obtain analysis results from the SKBR-3 cells. As a result, *PLK1* knock out using eCas12f1 occurred at an indel frequency of 36.69% (Fig. 5b), and we confirmed the presence of small deletions (Extended Data Fig. 8b). The clonogenic assay results for SKBR-3 cells treated with eCas12f1-*PLK1* showed an 81.82% reduction in colony numbers compared to the non-targeting control (NTC), validating that eCas12f1 efficiently reduced the survival rate of cancer cells by interfering with the reproductive ability (Fig. 5c,d).

This information also has been revised in line 373-396.

Minor points 1:

It is not always clear why certain genes are chosen to evaluate the editing efficiency of Cas12f1 variants. For example, gene editing efficiency of codon-optimized Cas12f1 and later Cas12f1_ge4.1 and Cas12f1_v1 are tested against a slightly different panel of genes (Suppl. Fig S1b,c) than Cas12f1_v1, Cas12f1_S1a and Cas12f1_S1b (Fig. 1b). The authors should state how and why a certain panel of target genes was chosen.

Response:

Thank you for pointing out the insufficient explanation regarding the selection of gene targets. There was no specific reason for the slight differences in the genes used in Fig. 1 and Extended Data Fig.1 (changed from Suppl Fig. S1). However, we agree with your

suggestion, and we conducted additional experiments to align the data in Fig. 1 and Extended Data Fig.1 to reflect the same set of genes. The reasons for gene selection are now provided in line 106-110.

Most of the gene targets used in this study are those previously used for engineering Cas12f1, and we have added explanations for the specific genes where applicable. In line 178-179, we mentioned the additional target gene, *PLK1*, a potential cancer therapy target for evaluating the indel efficiency of Cas12f1_mini variants, and in line 320-324, we described about the target gene selection for comparing the gene editing efficiency among the five CRISPR systems.

Minor points 2:

PLK1 is a well known therapeutic target in breast cancer and glioblastoma cells, whose knockout almost universally leads to cell cycle arrest and/or cell death. The authors use colony-formation assays to demonstrate reduced survival rate of U87 and SKBR-3 cells upon eCas12f1-induced *PLK1* knockout. Colony-forming assays, however, do not discriminate well between cell death and cell cycle arrest. The authors should either address this textually, or perform experiments to demonstrate cell death, apoptosis and/or cell cycle arrest upon *PLK1* knockout. Ideally, sorted, successfully transfected cells are used for such experiments.

Response:

Thank you for your suggestions that would further support the potential of eCas12f1 as a gene therapy tool for cancer. As you mentioned, *PLK1* is a well-known target in cancer therapy, and several studies have demonstrated that *PLK1* knockout using CRISPR significantly induces apoptosis in glioblastoma (reference27, PMID: 35442747), lung cancer cells (PMID: 38140096), and HeLa cells (PMID: 29686087).

We also investigated whether knockout using eCas12f1 actually induces cell death in SKBR-3, we stained the cells treated with either NTC or eCas12f1-*PLK1* using Annexin V, a calcium-dependent phospholipid-binding protein that binds to phosphatidylserine (PS) exposed on the outer membrane of apoptotic cells. We then counted the Annexin V-labeled cells using flow cytometry (Extended Data Fig. 8c). The results showed an 8.90% higher rate of apoptotic cells in the eCas12f1-*PLK1*-treated SKBR-3 cells compared to those treated with NTC, indicating that cell death was induced by

eCas12f1-mediated knockout (Extended Data Fig. 8c,d).

This information also has been added in line 390-396.

Minor points 3:

Knockout of PLK1 in U87 and SKBR-3 cells may lead to cell death in less than 72 hours. As such, the authors may underestimate the indel generation efficiency of eCas12f1, because negative selection of successfully edited cells may have occurred (Fig. 5g,h). This would also explain the rather large effect of PLK1 knockout on colony-forming potential of U87 cells, at an apparently low indel generation efficiency. It would be interesting to evaluate indel generation efficiency or PLK1 expression at an earlier timepoint.

Response:

Thank you for your valuable insights and interesting suggestions regarding our results and analysis. Following your suggestion, we investigated the indel efficiency of *PLK1* in HEK293T and SKBR-3 cell pools treated with eCas12f1 at time points before 72 hr.

The experimental results showed that the maximum indel efficiency was observed in HEK293T cells at 48 and 72 hr, while in SKBR-3, the indel efficiency increased over time, reaching its highest at 72 hr (bar graphs above). We speculated that negative selection due to *PLK1* knockout is not actively occurring within 72 hours based on the results of the accumulated eCas12f1 gene edits over time. This can be inferred from the presence of transcribed *PLK1* mRNAs and expressed *PLK1* proteins in the cells prior to eCas12f1 treatment. In the same context, the significant difference in cell growth between SKBR-3 cells treated with eCas12f1-*PLK1* and NTC observed in the colony formation assay performed after 14 days post-transfection indicates that by this time, the *PLK1*

molecules in the cells had nearly disappeared, making it sufficient time for the effects of eCas12f1-*PLK1* to become clearly evident as no further *PLK1* would be produced (Fig. 5c,d).

Minor points 4:

CRISPR is misspelled in line 351.

Response:

As you mentioned, we have corrected "CPRSPP" to "CRISPR" in the line 464. Thank you for pointing out the typo.

Reviewer #3

Response:

Thank you for informing us of your co-review as part of the Nature Communications initiative. We appreciate your and your co-reviewer's valuable feedback.

Reviewer #4

Compact CRISPR-Cas tools have great gene therapy potential due to their high-efficiency delivery capacity. UnCas12f1 is a previously discovered mini Cas enzyme. In this manuscript, the authors Park et al. optimized UnCas12f1 by improving gRNA and target DNA binding affinities in combination with previously identified mutations. The eCas12f1 shows high editing efficiency and is compatible with base editing and gene activation, which could be useful for the gene editing field. Before recommending it for publication, I have a few concerns.

Major points 1:

Off-target detection: the use of Cas-OFFinder with target sequencing may not provide comprehensive off-target identification. The authors may consider using de novo off-target identification methods such as GUIDE-seq, PEM-seq, CIRCLE-seq, or Digenome-seq for a more thorough off-target activity profile of eCas12f1.

Response:

We agree with your opinion. To achieve a more accurate and reliable off-target evaluation of eCas12f1, we have set up experiments for GUIDE-seq and analyzed off-target effects of eCas12f1 with two sgRNAs that showed high gene editing activity (*CRTAP*, *NLRC4*). The GUIDE-seq analysis revealed that no off-target sites were detected with eCas12f1 (Fig. 4e), suggesting that eCas12f1 offers specific and safe gene editing. This evidence further supports the potential of eCas12f1 as a valuable tool in the field of genetic engineering. This information also has been added in line 364-369.

Major points 2:

When comparing the editing efficiency of various CRISPR-Cas, the authors need to normalize the transfection rates. Cas9 and Cpf1 plasmids are larger in size compared to Cas12f1 tools, which may lead to lower levels of transfection.

Response:

First of all, thank you for your comments, which help us generate more accurate and reliable data in evaluating the gene editing activity of eCas12f1. We shared similar concerns, and to eliminate factors such as transfection efficiency that could affect the

gene editing efficiency of each CRISPR system, we considered three aspects to ensure a fair comparison of their gene editing activities.

First, we constructed a single vector system encoding both the Cas nuclease and sgRNA for each CRISPR system, thereby eliminating variables that could arise when using separate vectors. Second, to address the size differences in plasmids among CRISPR system, we used the same plasmid copy number (7.3×10^{10} copies) for transfection to focus solely on evaluating the cleavage activity of each CRISPR system. Since the plasmid sizes of each CRISPR system differ (9172 bp for SpCas9, 8937 bp for AsCpf1, 6743 bp for CasMINI_V3.1, 6152 bp for Cas12f1_ge4.1, and 6685 bp for eCas12f1), the molecular weight corresponding to the same copy number also varies (685.8 ng for SpCas9, 668.0 ng for AsCpf1, 504.2 ng for CasMINI_V3.1, 460.0 ng for Cas12f1_ge4.1, and 500 ng for eCas12f1). Therefore, we used the transfection reagent in the same ratio for DNA amount of each CRISPR system (1 μ l of P3000 reagent per 500 ng plasmid), according to the manufacturer's protocols. Finally, to minimize differences in transfection efficiency, we selected a highly efficient transfection method for HEK293T (Lipofectamine 3000). To assess any differences in transfection efficiency due to plasmid size, we counted GFP+ HEK293T cells treated with each CRISPR system after 72 hr post-transfection using flow cytometry. As shown in Extended Data Fig. 7a, we confirmed that the transfection efficiency of all CRISPR systems was sufficiently high. Furthermore, when comparing the gene editing efficiency from the total cell pool and only GFP+ cells across three genes, we found that the differences in gene editing efficiency among CRISPR systems analyzed in GFP+ cells are adequately reflected in the transfected cell pool as well (see the graphs below). Based on these findings, we concluded that it was reasonable to compare and analyze the gene editing activity of SpCas9, AsCpf1, and Cas12f1 variants using the entire cell pool, and we believed additional methods are not necessary in this case.

Minor points 1:

It's confusing to read the one-color figure panel in Fig. 2g, 2h, or similar ones. It's better to use a two-color system: >1 red, <1 blue.

Response:

Thank you for your comments, which help to make the differences between the data more apparent. As you suggested, we changed the highest fold change in gene editing efficiencies of the Cas12f1 variants to red and the lowest to light blue to enhance the visibility of the differences between the data (Fig. 2g,h).